



# A numerical study to investigate the roles of former hurricane Leslie, orography, and evaporative cooling in the 2018 Aude heavy precipitation event

Marc Mandement and Olivier Caumont

CNRM, Université de Toulouse, Météo-France, CNRS, Toulouse, France

**Correspondence:** Marc Mandement (marc.mandement@meteo.fr)

**Abstract.** In southeastern France, the Mediterranean coast is regularly affected by heavy precipitation events. On 14–15 October 2018, in the Aude department, a back-building quasi-stationary mesoscale convective system produced up to about 300 mm of rain in 11 h. The synoptic situation was perturbed by the former hurricane Leslie, involved in the formation of a Mediterranean surface low that focused the convective activity. At mesoscale, convective cells focused west of a quasi-stationary cold front and downwind of the terrain. To investigate the roles of Leslie, orography and evaporative cooling in the processes that led to the observed rainfall, numerical simulations are run and evaluated with near-surface analyses comprising standard and personal weather stations. Simulations show that, in a first part of the event, low-level conditionally unstable air parcels found inside strong updrafts mainly originate from the Mediterranean Sea, east of 4.5° E, whereas in a second part, an increasing number originates from Leslie's remnants. Air masses from east of 4.5° E appear as the first supplier of moisture over the entire event. Still, Leslie contributed to substantially moisten mid-levels over the Aude department, diminishing evaporation processes. Thus, the evaporative cooling over the Aude department does not play any substantial role in the stationarity of the cold front. Regarding lifting mechanisms, most of the air parcels found inside strong updrafts near the location of the maximum rainfall are lifted above the cold front, attesting its key role in focusing convection. Downwind of the Albera Massif, mountains bordering the Mediterranean Sea, cells formed by orographic lifting seem to be maintained by low-level leeward convergence, mountain lee waves and a favourable directional wind shear; when terrain is flattened, rainfall is substantially reduced. The location of the exceptional precipitation appears to be driven primarily by the location of the quasi-stationary cold front and secondarily by the location of convective bands downwind of the orography.

## 1 Introduction

Heavy precipitation events (HPEs), usually defined as events with daily rainfall exceeding 150 mm (Ricard et al., 2012), affect all the coastal areas of the western Mediterranean region, often producing flash floods (Nuissier et al., 2008). Due to the large societal impact of these events causing casualties and damage, they were extensively studied during the HyMeX programme extending from 2010 to 2020 (Drobinski et al., 2014; Ducrocq et al., 2014). Large rainfall amounts observed in time periods from few hours to several days during HPEs are the result of convective activity focusing over the same area. Strong convective activity often consists in continuous convective cell renewal constituting quasi-stationary mesoscale convective





systems (MCSs). The total rainfall amount observed over a given area is the product of both rainfall intensity produced by individual cells and the time period during which this convective activity remains focused over this area.

Ricard et al. (2012) built a climatology of HPE environments over the northwestern Mediterranean area. Synoptic situations favouring these HPEs over Languedoc-Roussillon, maritime part of the Occitanie region in southern France (Fig. 1), are now well known. At upper levels, a trough extends over the Iberian peninsula in a southeast-northwest orientation. This trough is as-

sociated with a cold low in the middle and high troposphere and generally entails a diffluent southwesterly flow at upper levels. MCSs develop preferentially northwards of a slow-evolving surface low located between Spain and the Balearic Islands focusing a southeasterly low-level jet (LLJ). The location and the deepening of the slow-evolving surface low has been identified as a key ingredient in focusing the convective activity over the same area and continuously initiating convective activity inside MCSs (Duffourg et al., 2016; Nuissier et al., 2016). The Mediterranean Sea supplies moisture and heat to this low-level airflow

through evaporation and heat exchange, which depend on the sea surface temperature (SST). Lebeaupin et al. (2006) showed that abnormally warm SST can destabilize atmospheric lower levels up to 2000–3000 m. In some HPEs, the Mediterranean Sea supplied up to 60 % of the total air parcels moisture (Duffourg and Ducrocq, 2013; Duffourg et al., 2018), modulating the intensity of convective precipitation. All these ingredients favour a persistent LLJ transporting low-level conditionally unstable air parcels over the Gulf of Lion. Several mechanisms are responsible for lifting the conditionally unstable low-level marine

flow when it reaches the coast, triggering convection over the same area. First, the mountainous terrain bordering the Mediterranean shore leads to orographic lifting. Secondly, in the lower levels of the atmosphere, low-level mesoscale boundaries, when stationary, can lift air parcels over the same areas. These mesoscale boundaries are found in fronts, outflow boundaries of cold pools, local convergence lines, mesoscale pressure troughs, among others. Lifting mechanisms include mechanical lifting in convergence areas and density departures between air masses that cause the most buoyant air parcels to ascend.

The orography-like lifting action of these mesoscale boundaries is combined with the action of the terrain itself (Ducrocq et al., 2008; Duffourg et al., 2018). This explains why large rainfall amounts are observed over the mountains as well as in the Mediterranean plains and over the sea. These mesoscale boundaries can pre-exist, or being initiated by the first convective cells. The idealized study of Bresson et al. (2012) shows that the occurrence of these mesoscale boundaries is dependent on the characteristics of the upstream flow: for example cold pools have been shown to form preferentially when the flow is relatively

dry or weak. Once initiated, the location and intensity of these features can be continuously modified by the MCS thanks to small-scale feedback mechanisms of the convection to the environment (Duffourg et al., 2016).

During the night of 14 to 15 October 2018, in the center and northwest of the Aude department (Fig. 1), part of Languedoc-Roussillon, rainfall accumulations over 200 mm in less than 12 h affected an approximately 60 km long and 10 km wide band oriented southeast to northwest. Inside the band, an automatic rain gauge in Trèbes (Fig. 2) measured 295.5 mm in 11 h in-

cluding 243.5 mm in 6 h and 110.5 mm in 2 h. Météo-France volunteer observers measured 318.9 mm in Conques-sur-Orbiel and 306.6 mm in Cuxac-Cabardès with manual rain gauges in 2 days, probably fallen almost entirely in 12 h as 93 to 99 % of the 2-day rainfall fell in 12 h in nearby Météo-France automatic rain gauges. In the center and northwestern part of the band, such 12 h rainfall accumulations were unprecedented in recent meteorological records and return periods were estimated over 100 yr. The orientation of the band, parallel to the small Trapel river catchment, led to a major flash flood in this catchment





in particular, overflowing and destroying bridges. It caused 15 fatalities, 75 injured and around 325 millions euros of damages including € 256 M to insurable assets for around 29 000 insurance claims and € 69 M to non-insurable assets (Préfecture de l'Aude, 2018; Ayphassorho et al., 2019; French Insurance Federation, 2019; Petrucci et al., 2020).

As described by Caumont et al. (2020) and Kreitz et al. (2020), at large scale, a remarkable and unusual feature of this event is the landfall of the former hurricane Leslie in the Portuguese coast on the evening of 13 October, one day before the Aude

HPE started. It is still unknown how the additional moisture provided by Leslie contributed to the convective system. At a smaller scale, largest rainfall accumulations are found to be aligned along bands downstream of the Pyrenees relief. Within the bands, the largest accumulations are found west of a quasi-stationary cold front and a quasi-stationary mesoscale trough. Because of the heavy rain observed in the area, evaporative cooling processes may have played a role in the stationarity of the cold front. Consequently, the goal of the article is to address the questions raised by Caumont et al. (2020): what was (i) the

origin of moisture including Leslie's contribution, (ii) the role of the Pyrenees relief and (iii) the role of the evaporative cooling in the physical processes that supplied conditionally unstable air, triggered convection and led to the observed rainfall?

Investigation of these questions is carried out as follows. First, the case study is presented in Sect. 2. Numerical simulations of this HPE produced with the Meso-NH model are described in Sect. 3. The realism of the reference simulation is evaluated near the surface through a comparison with independent analyses built from observations of standard and personal weather

stations in Sect. 4. Once evaluated, the simulation chosen as reference and a simulation to study the sensitivity to the terrain are used in Sect. 5 to investigate processes that led to the observed rainfall, in particular the role of Leslie. Then, the role of the cooling associated with the evaporation of precipitation is evaluated in Sect. 6.

## 2 Case description

The synoptic situation over Europe between 13 and 15 October was disturbed by the remnants of two Atlantic hurricanes,

Leslie and Michael, and their associated fronts (NOAA NESDIS, 2018).

At 500 hPa, on 13 October 12:00 UTC a large trough extended from the west of Iceland towards the west of Portugal (Fig. 3b). On 14 October 12:00 UTC (Fig. 3d), part of the trough split and evolved in a cut-off low over Spain, while the remnants of the former hurricane Michael generated a secondary trough at the rear of the cut-off. On 15 October 12:00 UTC (Fig. 3f), this secondary trough joined the existing cut-off low. The interaction between both lows, merged into a single cut-off

low, seems to have strongly slowed the westward movement of this mid-level cut-off low. Also, the cut-off low was prevented from moving westwards by the blocked situation observed over eastern Europe between 13 October 12:00 UTC and 15 October 12:00 UTC, due to a quasi-stationary high at all levels of the troposphere.

Between 14 October 12:00 UTC and 15 October 12:00 UTC, a small jet branch circumvented to the south the cut-off low. This jet branch reached maximum values over the Aude department of around $30\,\mathrm{m\,s^{-1}}$ at 300 hPa and around $25\,\mathrm{m\,s^{-1}}$ at

500 hPa. During this period, this jet showed some diffluence over the Aude department and changes were observed in its speed and direction due to the movement of the cut-off low. From southerly, the jet backed southeasterly, temporarily veered southwesterly at 300 hPa and then slowed down.



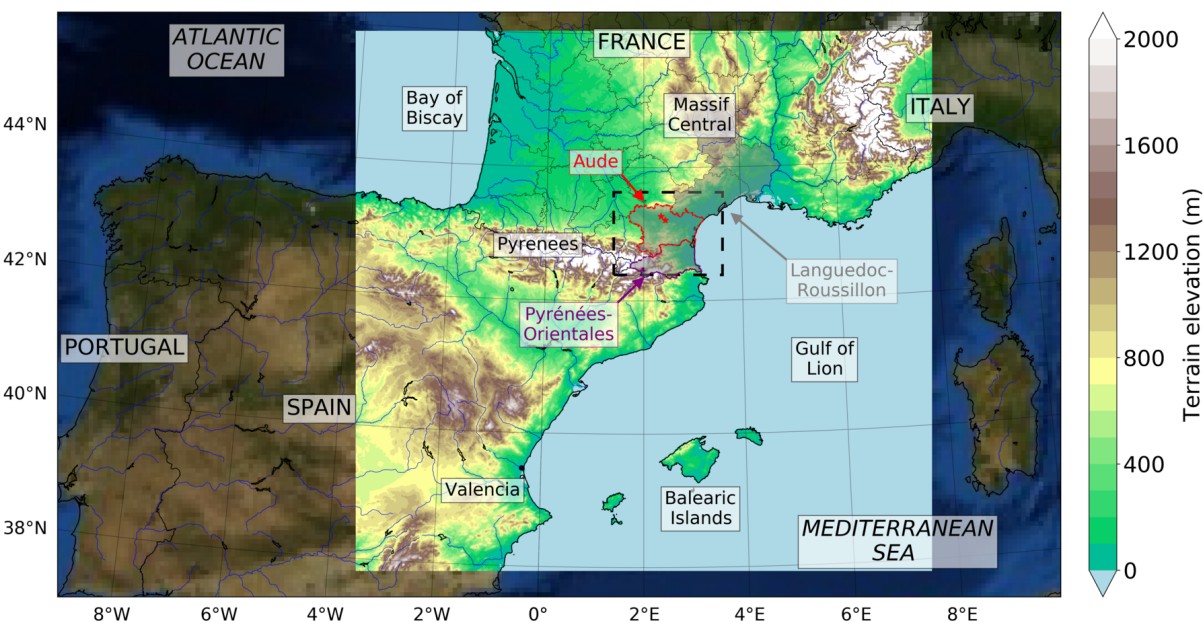

**Figure 1.** Map of southwestern Europe where locations mentioned in the article are indicated. The bright square and the dashed black line correspond to the two grid-nested domains (parent and child) of the simulation. Inside the bright square, terrain elevation from the parent model is shown; outside, it is the NASA visible blue marble image (from https://visibleearth.nasa.gov). Solid black lines indicate French departments and country borders. Languedoc-Roussillon, a region including Aude and Pyrénées-Orientales departments is shaded. The two little red stars indicate from north to south the location of Villegailhenc and Trèbes, two towns strongly affected by the HPE.

Near the surface, on 13 October 12:00 UTC (Fig. 3a), hurricane Leslie approached the Portuguese coast. A cold front along a mean sea level pressure (MSLP) trough linked Leslie to a low located over Ireland. At 15:00 UTC, few hours before
reaching the coast, Leslie was a category 1 hurricane with a 979 hPa estimated central MSLP (NOAA NWS National Hurricane Center, 2018a). Leslie was declared post-tropical cyclone at 21:00 UTC, just before landfall over Portugal between 21:00 and 22:00 UTC (NOAA NWS National Hurricane Center, 2018b). Leslie brought a large amount of moisture at all levels of the troposphere: over large areas, water vapour mixing ratio exceeded $12\,\mathrm{g\,kg^{-1}}$ at 925 hPa (Fig. 4a), $7\,\mathrm{g\,kg^{-1}}$ at 700 hPa and $2.5\,\mathrm{g\,kg^{-1}}$ at 500 hPa just before Leslie's landfall according to ARPEGE analyses. After landfall, Leslie's MSLP low filled up
quickly, and remnants of this former hurricane participated to extend towards south the existing cold front. On 14 October 12:00 UTC (Fig. 3c), the cold front has become more active over Spain and France. During the evening and the night of 14 to 15 October, a low rapidly deepened over the Mediterranean Sea, between the Balearic Islands and Valencia region around this cold front, associated with strong convective activity. A potential vorticity anomaly at upper levels is observed upstream of this low. It may have helped to deepen it through a baroclinic interaction, but its precise role would need further investigation.



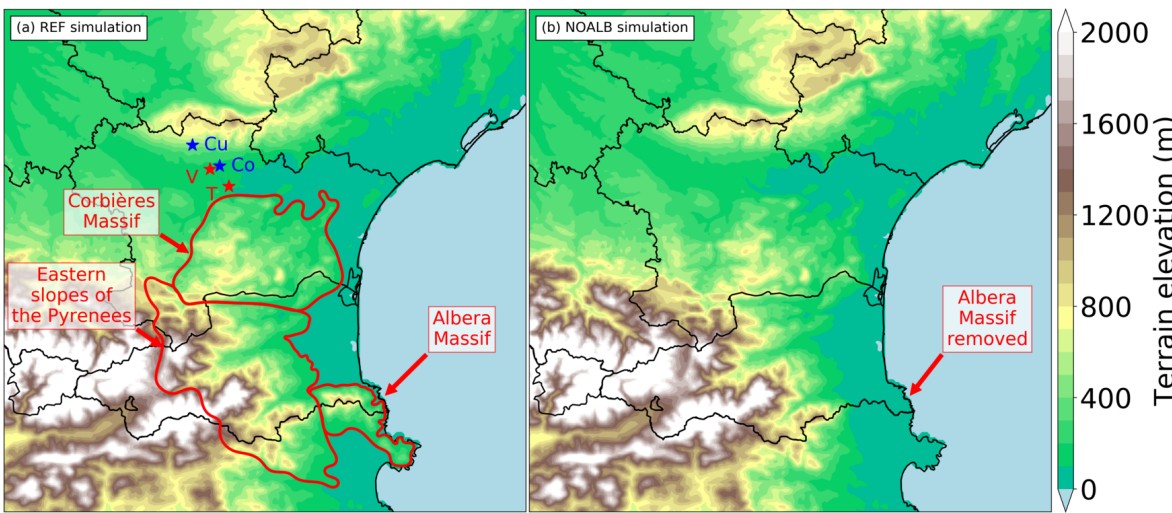

**Figure 2.** Orography of the south Languedoc-Roussillon including the Aude department in simulations (a) REF and (b) NOALB in which the Albera Massif is removed. Stars indicate towns affected by the HPE and mentioned in the article: "V" is Villegailhenc, "T" is Trèbes, "Cu" is Cuxac-Cabardès and "Co" is Conques-sur-Orbiel. The two red stars are landmarks displayed in other figures of the article. Red contours indicate mountain massifs.

This Mediterranean low moved slowly northwards overnight. It increased the MSLP gradient along the Languedoc-Roussillon coast and more generally over the Mediterranean Sea. Consequently, the near-surface east-southeasterly wind strengthened, increasing LLJ supply in warm and moist air originating from the Mediterranean Sea (dashed lines in Figs. 4d-f).

Caumont et al. (2020) show that a trough formed and extended this low towards Languedoc-Roussillon. At the same time, part of the active cold front (CF1) producing thunderstorms north of the Pyrenees (Fig. 3c) moved westwards. Its precipitating

activity decreased as it moved west, but its thermal signature near the surface remained. CF1 stopped in the middle of the Aude department, slightly west of the MSLP trough. Both CF1 and the MSLP trough remained quasi-stationary between 22:30 UTC 14 October and 04:00 UTC 15 October.

The advance northeastwards of the slow-moving low and its associated cold front (CF2, Figs. 3c,e) brought additional moisture over Languedoc-Roussillon. At low-levels, this moisture originates from both from Leslie's remnants and a moist

area already present over Spain (solid lines in Fig. 4). Kreitz et al. (2020) indicate that these different moisture contributions resulted in values of precipitable water up to 35 mm during the HPE. CF2 left the Aude department in the morning of the 15 October, which ended the HPE over the department.

A focus on the mesoscale situation shows that rain started over the Pyrénées-Orientales and Aude departments in the morning of 14 October. Météo-France standard weather stations in Pyrénées-Orientales recorded 0 to 20 mm rainfall, even 50 mm locally

near the Spanish border, rainfall amounts being higher over mountains, between 02:00 and 19:00 UTC 14 October. In Aude, 0 to 11 mm rainfall was recorded between 13:00 and 19:00 UTC. Observed rainfall was a share between convective cells locally





**Figure 3.** (a,c,e) Surface and (b,d,f) 500 hPa analyses of Météo-France national forecast department (Santurette and Joly, 2002) at (a,b) 12:00 UTC 13 October 2018, (c,d) 12:00 UTC 14 October 2018 and (e,f) 12:00 UTC 15 October 2018. Surface fronts are manually drawned using conventional observations, satellite, radar images and short-term forecasts (instead of analyses due to availability time constraints). Mean sea level pressure (in hPa), 500 hPa geopotential height (in gpm) and temperature (in °C) are from Météo-France operational global model ARPEGE (Courtier et al., 1991) 6 h forecast of the T−6 h run (T: time of the chart). Surface (respectively altitude) low-pressure centres are indicated by "D" (resp. "B") and high-pressure centres by "A" (resp. "H").





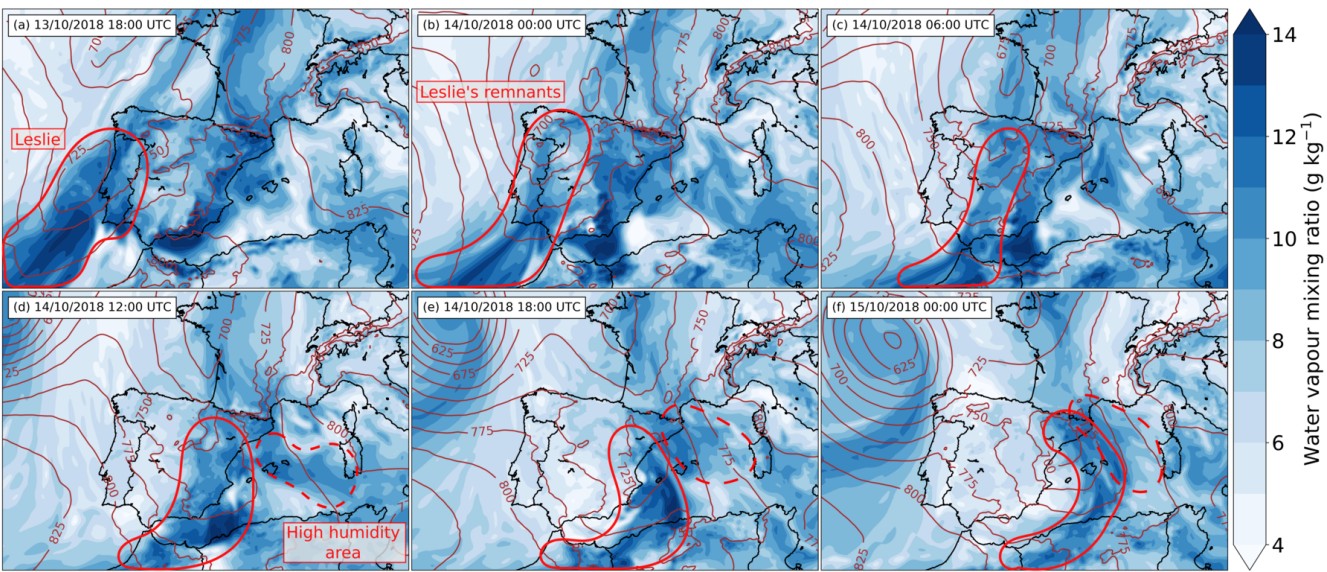

**Figure 4.** ARPEGE analyses of water vapour mixing ratio and geopotential height (in gpm) at 925 hPa on 13 October 2018 at (a) 18:00 UTC, on 14 October 2018 at (b) 00:00 UTC, (c) 06:00 UTC, (d) 12:00 UTC, (e) 18:00 UTC, and (f) on 15 October 2018 at 00:00 UTC. The approximate location of moist air masses carried by Leslie (solid red line) or found over the Mediterranean Sea (dashed red line) are circled.

formed and either convective cells or stratiform rain crossing the Pyrenees mountains from Spain. Clouds advected from Spain originated from a large cloud band ahead of CF2.

Over the Aude and Pyrénées-Orientales departments, the most intense rainfall occurred between 19:00 UTC 14 October and
07:00 UTC 15 October (Fig. 5). One shall note that the Opoul radar marked in Fig. 5a, well covering the area, had a failure between 21:55 UTC 14 October and 06:05 UTC 15 October. The three closest radars that filled the gaps are located to the northwest, north and northeast, outside of the area shown in Fig. 5. Thus, observed reflectivities are likely underestimated in Figs. 5b-d particularly around and south of the Opoul radar.

The first part of the HPE begins around 19:00 UTC 14 October. At that time, the size of the stratiform area and rainfall
intensity increased due to an increasing rain advection from Spain. East of this rainy area, near the Mediterranean coast, two parallel lines of convective cells formed, starting over the Albera Massif and the eastern slopes of the Pyrenees, and rapidly became the active parts of a back-building MCS. Fig. 5a exhibits these two parallel lines along the red arrows. The eastern line was more active than the western line: reflectivities exceed 40 dBZ at 21:35 UTC along the eastern line but remain below along the western line. At 23:00 UTC, Fig. 5b shows convective cores feeding this eastern line from its eastern flank with reflectivities
particularly strengthened in the northwestern part of the line. This organization in two lines is observed until around 00:00 UTC. After 00:00 UTC, the eastern line orientation slightly turned anticlockwise and a third active line formed, visible in Fig. 5c. Enhanced reflectivities along this line seems to originate in the eastern slopes of the Corbières Massif. Reflectivities of all lines are still strengthened in their northwestern parts. At 02:00 UTC, more reflectivities are observed over the Mediterranean Sea,



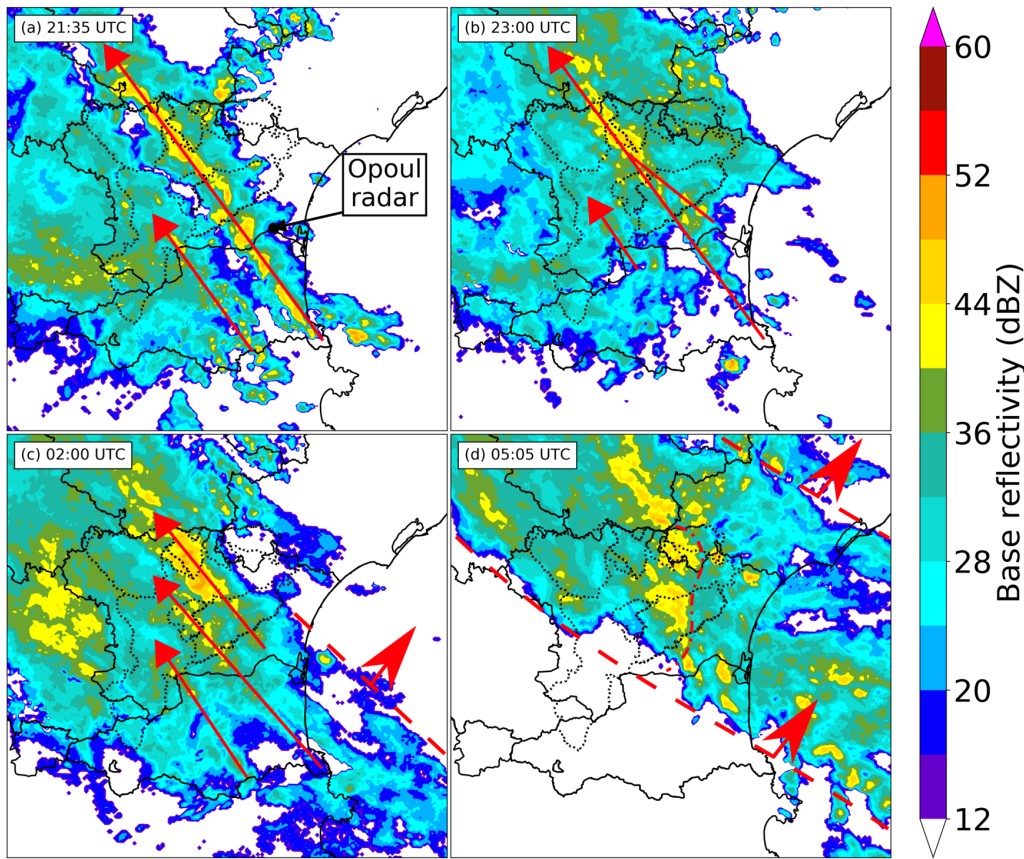

**Figure 5.** Radar base reflectivity from the French operational mosaic on 14 October at (a) 21:35 UTC, (b) 23:00 UTC and on 15 October at (c) 02:00 UTC and (d) 05:05 UTC. Red arrows with triangle arrowheads show the axes of continuous convective cell renewal and the movement of individual cells shown by radar. Dashed lines and large red arrowheads indicate the northeastwards movement of the convective system ahead of CF2.

south of the red dashed line in Fig. 5c, showing the advance of the convective system ahead of CF2. The large red arrows and

the dashed lines in Figs. 5c,d indicate the movement and the northeastern limit of this rain band, respectively. After 02:00 UTC, in the second part of the HPE, this rain band associated with CF2 modified the MCS organization in lines observed until then: active cells with reflectivity above 40 dBZ were continuously advected from the sea. Heavy rainfall persisted in the area, strengthened in particular west of the red dash-dotted line shown in Fig. 5d that corresponds to the location of CF1. After 05:00 UTC the large rain band associated with CF2 headed slowly northeastwards, leaving no more precipitation behind.

Fig. 6a shows the resulting 24 h accumulated precipitation from the ANTILOPE analyses, the Météo-France operational algorithm of quantitative precipitation estimation at a 1 km horizontal resolution, blending radar and rain gauge observations (Champeaux et al., 2009). In this case, available rain gauge observations from personal weather stations have been added to the product after a manual quality control, and validated by Caumont et al. (2020) over independent rain gauges. In ANTILOPE

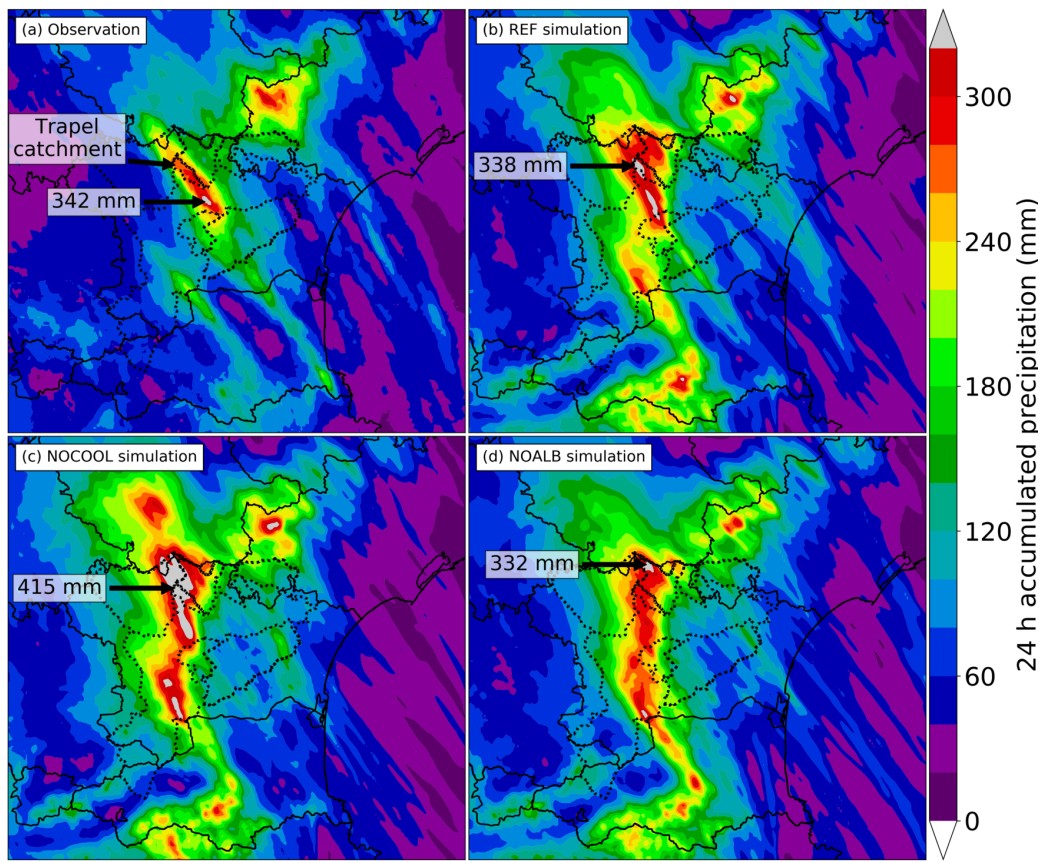

**Figure 6.** 24 h accumulated precipitation between 12:00 UTC 14 October and 12:00 UTC 15 October from the (a) ANTILOPE analyses, (b) REF, (c) NOCOOL and (d) NOALB simulations. Solid black lines indicate French departments and country borders. Dotted black lines indicate catchment limits of the Aude basin and its tributaries.

analyses, maximum accumulated precipitation reaches 342 mm few kilometres southwest of Trèbes, where an automatic rain

gauge measured 295.5 mm and close to a personal weather station that measured 311 mm. The line organization identified on radar observations resulted in two precipitation bands, referred to as eastern and western bands, with the eastern band a little curved, due to the formation of the third line between 00:00 and 02:00 UTC. The consequences were catastrophic near the observed precipitation maximum, because most of the rain has fallen in 6 to 12 h. Another local maximum of precipitation of 317 mm is observed over mountains, northeast of the bands. Consequences in this mountainous area were low since it is

accustomed to such rainfall and hourly precipitation accumulations remained moderate.





## 3 Numerical simulations

Simulations are performed with the French non-hydrostatic numerical research model Meso-NH version 5.4.2 (Lac et al., 2018), extensively used to study Mediterranean MCSs (Bouin et al., 2017; Martinet et al., 2017; Duffourg et al., 2018).

### 3.1 Meso-NH configuration

A two-way interactive grid nesting is chosen in order to study the sensitivity to a modification of model physics only within the child's model domain. A parent domain with a $960 \times 900 \, km^2$ horizontal domain over southern France and northwestern Mediterranean Sea is defined (Fig. 1). To realistically represent precipitating systems, a model configuration close to that of the French convection-permitting operational model AROME (Seity et al., 2011; Brousseau et al., 2016) is chosen with a 1 km horizontal resolution for the parent domain. The child's model horizontal domain is centered over the Aude department

with a $180 \times 135 \, km^2$ domain and a 500 m horizontal resolution. The Gal-Chen and Somerville (1975) height-based vertical coordinate is used with 89 stretched vertical levels from 5 m up to 23.75 km (height of the mass point), including 33 levels below 2 km height. The number and spacing of vertical levels is designed to be similar to AROME. A Rayleigh damping is progressively applied above 15 km height (i.e. the last 7 levels) to the perturbations of the wind components and the thermodynamical variables with respect to their large-scale values in order to prevent spurious reflections from the upper boundary. It

has a maximum value at the top of the upper absorbing layer of $0.001 \, s^{-1}$.

For the momentum transport scheme, a fourth-order centered discretization is used (CEN4H), while the transport scheme used for meteorological (temperature, water substances and turbulent kinetic energy) and scalar variables is a monotonic version of the piecewise parabolic method (PPM_01). The time integration scheme chosen is a fourth-order explicit Runge-Kutta (RKC4) and the model time step is 2 s for the parent, 1 s for the child. To suppress very short wavelength modes, a

fourth-order diffusion operator is applied to the wind components (u,v,w) with an e-folding time (time at which waves are damped by a factor $e^{-1}$) of 1800 s. Simulations start on 14 October at 12:00 UTC and last 24 h. The initial and lateral boundary conditions of the parent model are given by AROME operational analyses every 3 h at 1.3 km horizontal resolution. The lateral boundary conditions of the AROME analyses are provided by the latest available operational ARPEGE short cut-off analyses and forecasts (4 ARPEGE runs per day). For example, the analysis and 3-h forecast of the 06:00 UTC ARPEGE run provide

lateral boundary conditions for 06:00 and 09:00 UTC AROME analyses. There is one particular case for AROME 00:00 UTC analysis that receives its lateral boundary conditions from a very short cut-off ARPEGE 00:00 UTC analysis.

Earth surface variables and fluxes are simulated with the SURFEX model version 8.1 (Masson et al., 2013). Each grid mesh is divided in four main tiles. The following schemes are used for each tile: a three layer force-restore version of ISBA for natural land surface (Noilhan and Planton, 1989), TEB for urban area (Masson, 2000), the roughness length formula of

Charnock (1955) with Louis (1979) exchange coefficients for lake and the COARE 3.0 parametrization (Fairall et al., 2003) for sea-surface fluxes. COARE 3.0 is used as recommended by Rainaud et al. (2016) for Mediterranean HPEs and used by Lebeaupin Brossier et al. (2009) and Bouin et al. (2017). Optional corrections of sea-surface fluxes due to density effects during heat and water vapour transfer (Webb et al., 1980) and precipitation effects (Gosnell et al., 1995; Fairall et al., 1996) are





applied. The SST field comes from the initial AROME analysis and remains constant for the entire simulation. Physiographic
files used include the land cover data base ECOCLIMAP-II/Europe version 2.5 (Faroux et al., 2013), SRTM topography with
a horizontal grid spacing of approximately 250 m (Farr et al., 2007) and soil properties derived from harmonized world soil
database (HWSD; FAO, IIASA, ISRIC, ISS-CAS, JRC (2012)).

Regarding physical parametrizations, the longwave radiation scheme used is the Rapid Radiation Transfer Model (Mlawer
et al., 1997) while the shortwave scheme is based on Fouquart and Bonnel (1980) method. Full radiation computations are
performed once every 15 min. For turbulence, the one dimensional parametrization used is based on a 1.5-order closure (Cuxart
et al., 2000) of the turbulent kinetic energy equation with the Bougeault and Lacarrere (1989) mixing length. At 1 km resolution,
deep convection is assumed to be resolved explicitly by the model's dynamics. Shallow convection is parametrized with the
Pergaud et al. (2009) Eddy Diffusivity Mass Flux scheme. The bulk one-moment mixed microphysical scheme used is ICE3
(Pinty and Jabouille, 1998; Caniaux et al., 1994) that includes six water species (water vapour, cloud droplets, raindrops,
pristine ice crystals, snow or aggregates and graupel).

For the child model at 500 m horizontal resolution, in the so-called turbulence "grey zone", the choice of a 1D turbulence
parametrization can be questioned. This choice was made to keep consistency between the two coupled models. Investigations
of Machado and Chaboureau (2015) showed that 1D turbulence parametrization produces too many small cloud systems and
rain cells with a shorter lifespan than observed. Although very sensitive to the in-cloud mixing length chosen, 3D turbulence
parametrization appears more consistent with observations. Further investigations, out of the scope of the study, are required on
Mediterranean cases to compare 1D and 3D turbulence parametrizations at 500 m resolution. Preliminary investigations were
done by Martinet et al. (2017) showing some improvements by using a 3D turbulence parametrization with 3 different mixing
lengths in simulations of a Mediterranean HPE at 500 m horizontal resolution compared to a 2.5 km resolution simulation using
a 1D turbulence parametrization. Because of the difference in horizontal resolution, it is difficult to conclude what benefit came
from the highest resolution or from the turbulence scheme.

Regarding computation setup, each experiment was computed on the Météo-France supercomputer Bullx DLC b710 "Beau-
fix" with 3200 cores (40 cores by node for 80 nodes) in approximately 14 h. The Meso-NH model was compiled at an opti-
mization level 2 (O2) in order to produce bit reproducible results.

## 3.2 Experiments

Three simulations are shown in the study. The first one is run to realistically simulate the case. Validation of its realism is
performed in Sect. 4. This reference study is called REF hereafter.

The importance of mountains bordering the Mediterranean shore in focusing or enhancing convection was pointed out by
Ducrocq et al. (2008) in the 1999 Aude HPE that affected an area 30 km east of the one affected in the 2018 Aude HPE. The
authors hypothesized that the Massif Central and the Pyrenees probably enhanced the intensity of the rainfall by channelling
the warm and moist LLJ. Removing the Massif Central diminished the maximum precipitation simulated though it had not a
substantial impact on the stationarity of the simulated MCS, organized along a narrow line (Nuissier et al., 2008). However the
removal of Pyrenees mountains, located upstream of the precipitation maximum, was not tested.




In the 2018 Aude HPE, radar observations in Figs. 5a-c show that a large number of convective cells were continuously initiated during several hours over the Albera Massif and remained aligned downstream. The Albera Massif is the easternmost Massif of the Pyrenees bordering the Mediterranean Sea (Fig. 2); its highest mountain is the Neulos peak which culminates at an altitude of 1256 m. Its altitude is 1023 m (respectively 1128 m) in the father (resp. child) model. These observations support the occurrence of quasi-stationary convective banding that can lead to large rainfall accumulations. Similar convective band generation was observed in the southeastern flank of the Massif Central by Miniscloux et al. (2001) and Cosma et al. (2002) upstream of small-scale topography ridges, with an enhancement of these bands on the lee side of the ridges. Cosma et al. (2002) showed in both idealized and real-case simulations that the extension of precipitation lines downwind of orography resulted (i) from the formation of a mountain wave immediately downwind of the crest and (ii) from the lee-side convergence created by deflection around the obstacle. Sensitivity tests indicated that the structure (length, width) and the intensity of the rain band are quite dependent on the upwind meteorological conditions and on the topographic configuration. The strengthening of convection downstream of small-scale topographic structures due to lee-side convergence was also noted by Ricard (2005) during the 1995 Cévennes HPE. Barrett et al. (2015) attributed a similar event over the UK to lee-side convergence combined with thermally forced convergence resulting from elevated heating over the upstream terrain.

Thus, to clarify the effect of the Albera Massif on precipitation, a simulation replacing it by a flat terrain, called NOALB, is carried out. Inside the Albera Massif area bounded by the red solid line (Fig. 2a) terrain elevations above 25 m a.s.l. are set to 25 m a.s.l., except west of the area in order to avoid the abrupt transition to higher terrain. From the western limit of the area, terrain elevations are set to gradually decrease at a rate of 40 m every km eastwards (4 % slope) until reaching 25 m a.s.l. Such transition is not necessary north or south of the Albera Massif because terrain elevation is mainly below 25 m. The topography resulting from these changes is shown in Fig. 2b.

In previous HPEs such as the 2002 Gard case or the HyMeX IOP13 case, evaporation processes have been shown to play an important role in cooling the air near the ground (Ducrocq et al., 2008; Duffourg et al., 2018). In these previous cases, dry air parcels at altitudes between 1000 m and 4000 m, when mixed to precipitation, were humidified and cooled through evaporation processes, forming vigourous downdraughts resulting in the formation and maintenance of cold pools. In the 2018 Aude HPE, the role of evaporative cooling in modifying the location of CF1, changing the duration of CF1 stationarity or substantially cooling the cold sector west of CF1 is questioned. To clarify the effect of the cooling associated with the evaporation of precipitation on the case study, this process is switched off for the child model (black dashed square in Fig. 1) but kept for the parent model (bright square in Fig. 1) in a simulation called NOCOOL. Because of the two-way setup chosen, changes applied to the child model interact with the parent model. This change is applied only to the child model because applying these modifications to the entire parent model strongly modifies MSLP and wind fields upstream of Languedoc-Roussillon, probably because cooling processes have a strong influence on the life cycle of the low over the Mediterranean Sea (not shown). Applying these modifications only to the inner domain allows us to quantify the impact of this process only over the south of Languedoc-Roussillon where strongest precipitation is observed.





## 4   Validation of the REF simulation near the ground

Near-surface fields of the REF simulation are compared to independent gridded analyses built from screen-level observations of standard and personal weather stations, called SPWS analyses, described by Mandement and Caumont (2020). Instead of directly comparing the two fields, to separate physical departures from departures due to gridding methods, REF is interpolated

in the same way as the SPWS analyses. This interpolation is called REF_SP. The method consists in replacing the value and altitude of each weather station used in the SPWS analyses by the value and altitude of REF nearest grid point. Then, with the exact same weights and gridding method used in the SPWS analyses, the REF_SP gridded field is built. The method gives information about model features that are not resolved by the observation network used in the SPWS analyses. Thus, in this section, the REF_SP interpolations are compared to the SPWS analyses (the REF fields are also shown to illustrate the

method). For precipitation fields, REF is directly compared to the ANTILOPE analyses because they are on regular grid whose resolution is close to that of the model.

Regarding rainfall accumulations, the REF simulation is able to reproduce the organization of precipitation in two bands oriented southeast to northwest (Figs. 6a,b) and located approximately as the observations. Local maxima of both bands are also located quite correctly. Along the eastern band, two local precipitation maxima are simulated: a first one north of the Trapel

catchment with 338 mm and another one with 331 mm at the same latitude but 7.3 km west of the observed 342 mm maximum. Rainfall along the southern part of this eastern band is underestimated. Along the western band, the local maximum is largely overestimated with 296 mm whereas the observation estimates 206 mm and in the southern part of the band, simulated rainfall amounts reach more than twice the observations. In the upper right corner of Fig. 6b, the shape of the area affected by heavy rainfall and the maximum accumulated rainfall simulated (343 mm) are similar to observations (317 mm observed). Elsewhere,

the REF simulation generally overestimates rainfall, particularly over orography. These overestimations are substantial north and east of the local maximum indicated by the black arrow (Fig. 6b) or over the Pyrenees mountains.

Regarding the timing of rainfall, the first part of the HPE begins around 19:00 UTC 14 October both in REF and in observations (see Sect. 2). Until 23:00 UTC, rainfall simulated is less intense than observed particularly along the eastern band: north of it, hourly rainfall above 20 mm is observed after 21:00 UTC but simulated only after 23:00 UTC. The REF simulation is able

to reproduce the convective lines that are observed mostly in the first part of the HPE but substantial rain rates along these lines are simulated later than in observations. The second part of the event begins around 05:00 UTC in the REF simulation whereas it is 02:00 UTC in observations. Finally, the HPE ended when the rain band associated with the cold front CF2 left the Aude department. It left the western part of the Aude department (respectively the entire Aude department) heading northeastwards around 10:00 UTC (resp. 10:30 UTC) in the REF simulation, while it was observed around 07:00 UTC (resp. 09:00 UTC). So,

it rained longer over the Aude department in the REF simulation than in observations, particularly in the western part of the department.

As described in Sect. 2, Caumont et al. (2020) show that the location of a MSLP trough and a virtual potential temperature ($\theta_v$) gradient remained quasi-stationary between 22:30 UTC 14 October and 04:00 UTC 15 October. The ability of REF to accurately model these two features is evaluated.



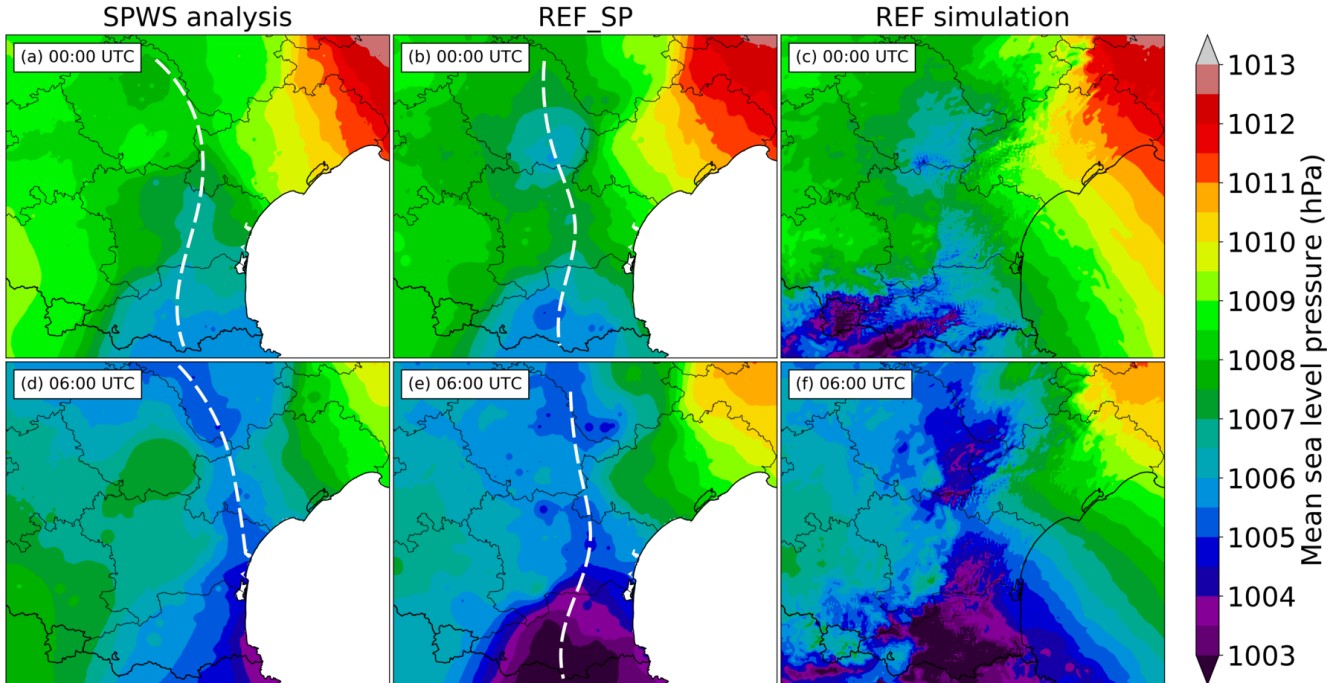

**Figure 7.** Mean sea level pressure on 15 October at (a,b,c) 00:00 UTC and (d,e,f) 06:00 UTC from (a,d) the SPWS analysis, (b,e) REF_SP and (c,f) the REF simulation. Solid black lines indicate French departments and country borders. Dashed white lines indicate the approximate location of the MSLP trough. Analyses are not computed over the Mediterranean Sea, in white, because of the lack of surface observations over sea.

Regarding MSLP, at 00:00 UTC, both the SPWS analysis (Fig. 7a) and REF_SP (Fig. 7b) show the MSLP trough. REF_SP locates the trough slightly west compared to the analysis. MSLP is up to 3 hPa higher in the northern part of the trough in analyses compared to REF_SP. At 06:00 UTC, analysis (Fig. 7d) shows that the MSLP trough moved 30 km eastwards whereas the REF simulation (Fig. 7e) shows that the trough barely moved since 00:00 UTC. In both cases, the trough deepened between 00:00 UTC and 06:00 UTC. REF_SP keeps the trough at a quasi-stationary location until about 07:00 UTC whereas

analyses indicate the trough remained quasi-stationary only until about 04:00 UTC, before moving eastwards. The comparison between REF_SP and the SPWS analysis reveals a time lag of approximately 3 h in the movement of the trough in the morning of 15 October.

    Regarding $\theta_v$ at screen level, at 23:00 UTC, SPWS analysis (Fig. 8a) locates in the middle of the Aude department the $\theta_v$ gradient which is a salient feature of the cold front CF1. In the analysis, $\theta_v$ is below 18 °C west of CF1 and above 20 °C east

of it. REF_SP (Fig. 8b) shows a similar amplitude of $\theta_v$ gradient at the approximate same location as the SPWS analysis but $\theta_v$ is 1 °C higher on both sides of CF1. This shows that the location of CF1 given by REF simulation (Fig. 8c), slightly more west of what is seen in Figs. 8a,b, is consistent with the SPWS analysis. The sharp east-west $\theta_v$ gradient is also consistent



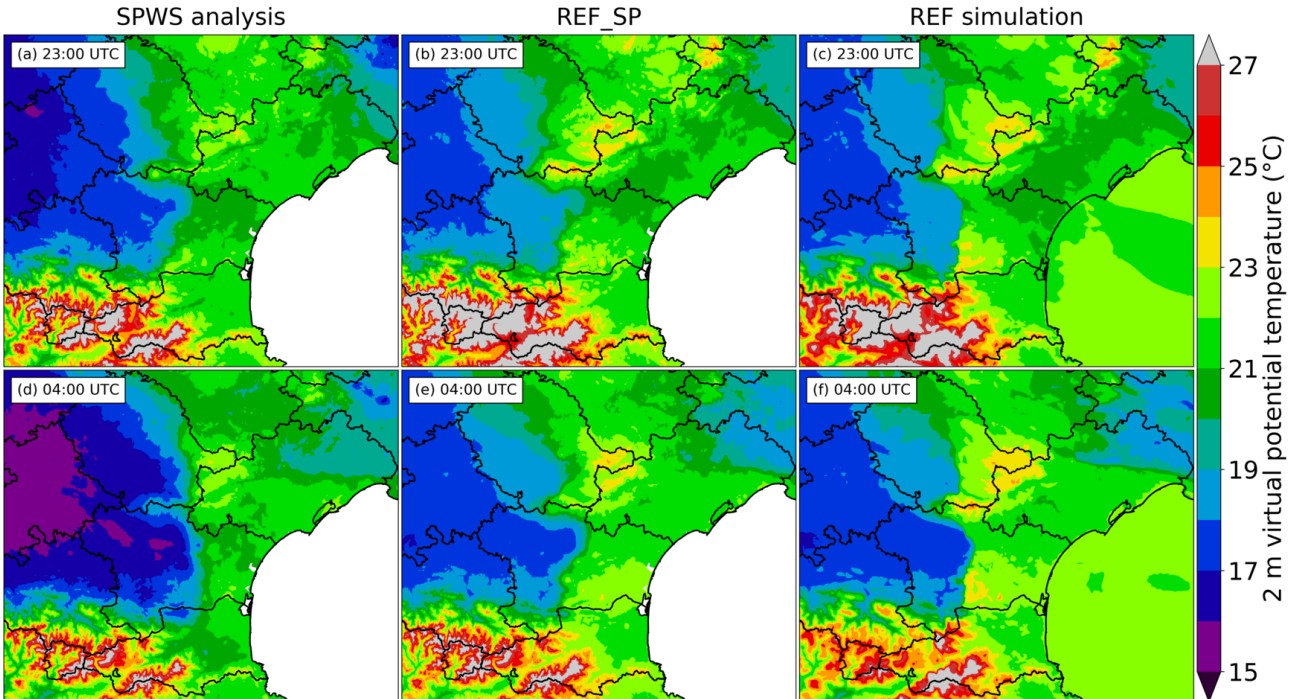

**Figure 8.** 2 m height virtual potential temperature at (a,b,c) 23:00 UTC 14 October and at (d,e,f) 04:00 UTC 15 October from (a,d) the SPWS analysis, (b,e) REF_SP and (c,f) the REF simulation.

with the observations even if REF overestimates by approximately 1 °C the virtual potential temperature on both sides of CF1. In fact, the comparison between Figs. 8a,b,c shows that the surface observation density is unable to seize the precise location

of $\theta_v$ gradient such as the one given by REF. At 04:00 UTC, $\theta_v$ decreased by as much as 2 °C west of CF1 in the SPWS analysis (Fig. 8d) and decreased by less than 1 °C in REF_SP (Fig. 8e) compared to the state at 23:00 UTC. This decrease is essentially due to a temperature decrease, relative humidity remaining between 90 and 100 % at 23:00 and 04:00 UTC. In the SPWS analysis, south of the Aude department, CF1 moved eastwards: such displacement is not observed in REF simulation (Figs. 8e,f).

To finely describe the movement of CF1, the location of the 19 °C $\theta_v$ isotherm is shown in Fig. 9. SPWS analysis (Fig. 9a) shows that CF1 moved eastwards between 19:00 and 22:00 UTC. It remained quasi-stationary east of the two red stars between 22:30 and 04:00 UTC. After that time, it moved eastwards and reached the Mediterranean Sea around 06:30 UTC. REF_SP (Fig. 9b) shows that CF1 became quasi-stationary earlier, around 21:00 UTC, and remained quasi-stationary until 07:00 UTC. It reached the Mediterranean Sea around 09:30 UTC. Thus, CF1 is quasi-stationary approximately 4.5 h longer in REF than in

analyses and reaches the Mediterranean Sea with a 3 h delay in REF compared to analyses. Regarding the location where CF1 remained quasi-stationary, setting aside delays, REF_SP generally locates CF1 slightly more west than the SPWS analyses, but the westwards shift remains mostly below 10 km. Comparison between REF_SP and REF (Figs. 9b,c) shows some small-scale

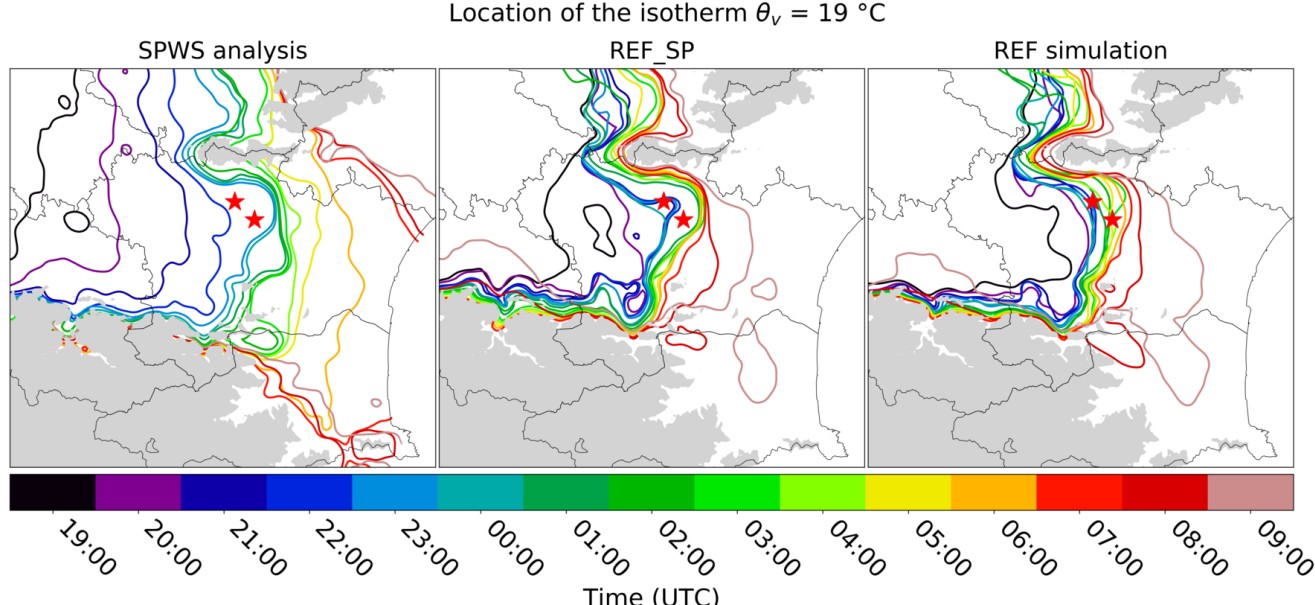

**Figure 9.** Location of the 19 °C virtual potential temperature ($\theta_v$) isotherm at 2 m height between 19:00 UTC 14 October and 09:00 UTC 15 October from (a) the SPWS analysis, (b) REF_SP and (c) the REF simulation. Terrain elevation over 750 m above sea level (a.s.l.) is shaded in grey. Red stars show the location of Villegailhenc (northern star) and Trèbes (southern star), towns strongly damaged by the HPE.

movements of CF1 simulated by REF that cannot be reproduced in REF_SP. It gives insight about the magnitude of departures that cannot be seized by the SPWS analyses and thus should not be considered as substantial.

Fig. 10 shows 10 m height horizontal wind simulated by REF with surface observations superimposed. At 23:00 UTC, REF simulates 12 to 16 m s$^{-1}$ southeasterly to east-southeasterly winds blowing over the Mediterranean Sea near the Languedoc-Roussillon shore. Inland, east of CF1, east-southeasterly winds reach mostly 6 to 12 m s$^{-1}$. West of CF1, westerly 0 to 6 m s$^{-1}$ winds are observed. These opposite winds cause strong wind convergence at CF1 location. Small differences are identified between REF and observations around CF1: they support that CF1 location is 0 to 10 km further west in the REF simulation than

in the SPWS analyses. Inland, between CF1 and the coast, observations support eastern wind direction more than southeastern direction and indicate slight underestimations. At 04:00 UTC, as a result of the approaching Mediterranean low, REF wind speed increased east of CF1, reaching 14 to 18 m s$^{-1}$ over the Mediterranean Sea and 8 to 14 m s$^{-1}$ inland. At that time, 3 stations indicate westerly winds while REF indicates the opposite, showing that REF simulates CF1 location 10 to 15 km further west than observed.

In summary, the REF simulation produces realistic near-surface fields in comparison with the SPWS analyses and scattered wind observations. The main differences between analyses and REF are substantial time lags in the stationarity of mesoscale boundaries: the MSLP trough and CF1 remained quasi-stationary between 3 to 4.5 h longer over the Aude department in the REF simulation than in the SPWS analyses. After 04:00 UTC, it results in an approximately 3 h delay in the movement of



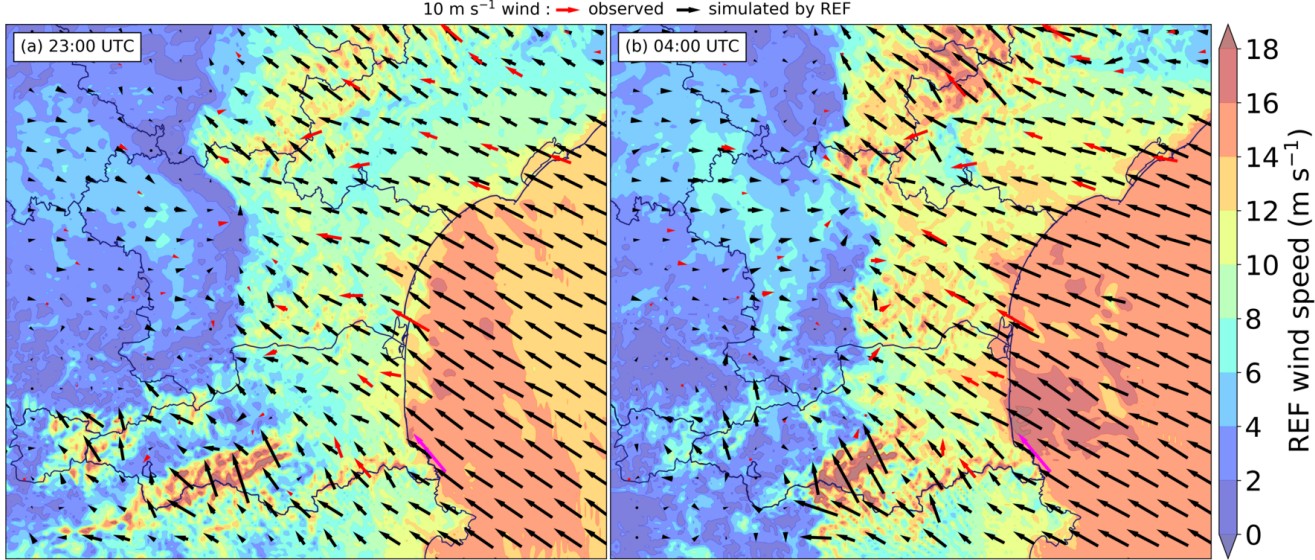

**Figure 10.** 10 m height instantaneous horizontal wind speed and direction simulated by REF (black arrows, speed also displayed with filled contours) and 10 m height 10 min averaged horizontal wind speed and direction observed (red arrows) at (a) 23:00 UTC 14 October and at (b) 04:00 UTC 15 October. The arrow length is proportional to the wind speed. Purple arrow is the Cap Béar observation measured at an unconventional height of 36 m.

these mesoscale boundaries in REF compared to analyses. This longer stationary period probably caused the prolonged rainfall
over the area that may explain some of the overestimations found. Also, the westerly shift found between REF and the SPWS analyses in the quasi-stationary location of CF1 or the MSLP trough is correlated with the westerly shift found in the location of the heaviest precipitation between REF and the ANTILOPE analysis. The amplitude of both shifts is about 10 km. Keeping in mind these departures, the REF simulation is considered realistic enough to study the case and is taken as the reference in the following sections.

## 340  5  Origin of the conditionally unstable air and lifting mechanisms

This section investigates what mechanisms supplied conditionally unstable air to the convective system, including studying whether its moisture comes from areas that have been particularly humidified by Leslie's remnants. Trajectories and thermodynamic properties of air parcels that contributed to the formation of the strongest convective cells in the REF simulation as well as those of cold parcels located west of CF1 are described. Lifting mechanisms are also studied, in particular the role of the
Albera Massif. To carry out the investigation, a series of backward trajectories are computed using the Lagrangian trajectory tool of Gheusi and Stein (2002). Atmospheric columns with strong mid-level updrafts close to the location of the 24 h maximum rainfall are selected. Inside each column, 40 air parcels taken every 2 vertical levels of the model from the second level





(18 m height) to the 80[th] (13 568 m height) are followed and their trajectories are shown in the figures. Some trajectories (in grey in Fig. 11a) are not projected on the vertical sections when vertical movements of the air parcels are of small amplitude around the Aude department; or when a trajectory intersects the terrain along the projection axis because the terrain along the axis differs substantially from the one along the trajectory.

## 5.1 First part of the HPE

This part lasts approximately between 19:00 UTC 14 October and 05:00 UTC 15 October in the REF simulation.

At 00:00 UTC 15 October, air parcels below 10 km found in the atmospheric column "D" (Fig. 11a) where a convective updraft is simulated originate from 3 preferential directions.

Air parcels under 800 m a.s.l. are inside the cold sector west of CF1, where virtual potential temperature is below 21 °C (Fig. 11b). They originate from the west, mainly the southeast of the Bay of Biscay and the north of Spain. They followed the same trajectory as CF1 that was located there at 12:00 UTC 14 October (see Fig. 3c). They remained most of the time below 500 m a.s.l., and some were slightly lifted near CF1 boundary.

Parcels found above 800 m a.s.l. in "D" are ascending air parcels (Fig. 11c). These parcels originate from east and northeast of the Balearic Islands, over the Mediterranean Sea, away from the convective system ahead of CF2 located between the Balearic Islands and the Pyrenees at 00:00 UTC (Fig. 12c). During their transport over the Mediterranean Sea, the altitude of these parcels remained almost constant between 0 and 1.4 km a.s.l. When they reached the coast, some experienced slight lifting over the Albera Massif and the Corbières Massif but remained below 2 km a.s.l. They were finally lifted up to 6.5 km a.s.l. above CF1. Parcels are carried by the marine LLJ shown in Fig. 12b. At 697 m height, the speed of this southeastern LLJ exceeds 20 m s$^{-1}$ between the east of the Balearic Islands and the Aude department, up to 26 m s$^{-1}$ near the Languedoc-Roussillon shore and inland. Such a wind speed transports quickly these air parcels: they travelled from B to D (Fig. 11) in approximately 12 h. Over the Aude department, large wind convergence is simulated: horizontal wind speed brutally decreases from 26 to near 0 m s$^{-1}$, leading to enhanced ascending movements along CF1.

These parcels originate from moist areas over the Mediterranean Sea (dashed lines in Fig. 4b): at 14:00 UTC 14 October, the 19 air parcels shown in Fig. 11c already had a mean water vapour mixing ratio of 9.1 g kg$^{-1}$ (Tab. 1). Their water vapour mixing ratio increased through their transport above the Mediterranean Sea by 1.3 g kg$^{-1}$, reaching 10.4 g kg$^{-1}$ at 20:00 UTC 14 October. During their lifting, between 22:00 and 00:00 UTC, they released moisture: their water vapour mixing ratio decreased by an average of 4.0 g kg$^{-1}$. Some of this moisture is released through condensation processes inside the convective clouds, before some of the water condensates eventually precipitate. The high moisture and relatively warm temperatures in the area of origin of these parcels is shown in Fig. 12a by equivalent potential temperatures above 53 °C at 697 m height. Consequently, air parcels carried by the LLJ are conditionally unstable: simulated most unstable convective available potential energy (MU-CAPE) reaches 100 to 600 J kg$^{-1}$ over Languedoc-Roussillon and 600 to 1200 J kg$^{-1}$ over sea. The 3D CAPE field (not shown) shows that highest CAPE values are mostly reached at the first model level (5 m height), and decrease rapidly with height.

The MSLP low that drives the LLJ is located in the area of light winds over Spain shown in Fig. 12b. Northeast and east of it, between the Balearic Islands and the Pyrenees, strong convective cells are triggered ahead of CF2. Several convective





**Figure 11.** (a) Horizontal projection of the 40 backward trajectories from air parcels taken inside the atmospheric column located in "D" (43.25° N, 2.25° E) at 00:00 UTC 15 October in the REF simulation. Trajectories in colour are the ones projected in (b-d): their colour varies according to the water vapour mixing ratio of the parcels. Other trajectories are in grey. Trajectories are computed until 12:00 UTC 14 October, except for parcels that reach domain boundaries before that time. (b-d) Vertical projections of backward trajectories along the dashed black lines shown in (a) and corresponding cross sections of virtual potential temperature at 00:00 UTC 15 October. Each parcel is projected on the section closest to its trajectory inland. Terrain is in black.







**Figure 12.** REF simulation at 00:00 UTC 15 October of (a) equivalent potential temperature ($\theta_e$) at 697 m height, (b) wind at 697 m height, (c) most unstable CAPE with instantaneous precipitation rate and (d) wind at 2957 m height.

cells are advected by the south-southeasterly mid-level wind towards Languedoc-Roussillon. Thus, in Fig. 11d some parcels above 7 km a.s.l. originate from the Mediterranean low and its associated front CF2. These parcels, from altitudes between 1.8 and 5.5 km, have lower water vapour mixing ratios than parcels carried by the LLJ. Some of these parcels are lifted over the 385 Pyrenees mountains and one is lifted over CF1 because it managed to cross the Pyrenees without being lifted.

REF simulates southeasterly low-level winds (Fig. 12b at 697 m height) and south-southeasterly mid-level winds (Fig. 12d at 2957 m height), showing directional wind shear in the lower part of the troposphere. The MSLP trough simulated over the Aude department may locally increase this wind shear inland: Fig. 10a shows that the LLJ backs from the southeast to the east-southeast inland. Backward trajectories confirm it: the higher the air parcel comes, the further west it originates (Fig. 11a).





**Figure 13.** As Fig. 11 for trajectories ending at 04:00 UTC 15 October at 43.31° N, 2.28° E.

**Table 1.** Mean water vapour mixing ratio (g kg$^{-1}$) of air parcels originating from below 1500 m height along axes shown in Figs. 11,13,14 as a function of time. Dash indicates that at least one parcel was out of the simulated domain at that time.

| Trajectory end (UTC) | Axis | Number of parcels | Time (UTC) | | | | | | | | |
|---|---|---|---|---|---|---|---|---|---|---|---|
| | | | 14:00 | 16:00 | 18:00 | 20:00 | 22:00 | 00:00 | 02:00 | 04:00 | 07:00 |
| 00:00 | Fig. 11c: B → D | 19 | 9.1 | 9.6 | 9.9 | 10.4 | 10.1 | 6.1 | | | |
| 04:00 | Fig. 13b: A → D | 8 | – | – | – | 10.5 | 10.8 | 11.0 | 11.0 | 5.7 | |
| 07:00 | Fig. 14b: A → B | 7 | – | – | – | – | – | 8.6 | 9.8 | 10.4 | 5.3 |





At 04:00 UTC 15 October, backward trajectories in Fig. 13 picture a situation almost identical to the one from 00:00 UTC. Most of the parcels lifted in the selected updraft are still supplied by the LLJ, coming from altitudes between 0 and 1.2 km for parcels projected along the axis A–D (Fig. 13b) and between 0.8 and 1.5 km for parcels along the axis B–D (Fig. 13c). Parcels projected along the axis A–D, coming from the lowest levels supplied large amounts of moisture, slightly increasing during their transport from 10.5 to 11.0 g kg$^{-1}$ between 20:00 and 02:00 UTC (Tab. 1). These parcels were lifted up to 7 km a.s.l.

immediately above CF1. Most of the air parcels along the axis B–D had a similar path and were lifted up to 8 km a.s.l., except two parcels that were lifted above the orography near 650 km along B–D. Only few elevated parcels came from the Balearic Islands, i.e. from inside the convective system ahead CF2. They were lifted above the Pyrenees (Fig. 13d) and reached levels above 9 km a.s.l. in column "D" with a water vapour mixing ratio before they reached the column lower than 4 g kg$^{-1}$.

## 5.2   Second part and end of the HPE

This part lasts approximately between 05:00 and 10:00 UTC 15 October in the REF simulation. Around 05:00 UTC, the cold front CF2 reached the Pyrénées-Orientales coast. Contrary to the first part during which most of convection is triggered inland, in this second part, convection is triggered over the Mediterranean Sea and carried inland by the mid-level wind, diminishing the influence of local forcings in triggering convection. Convection triggered over sea is fed by a warm and moist air mass ahead of CF2 with equivalent potential temperature up to 59 °C (Fig. 15a). Consequently, this air mass is more unstable than

in the first part: MUCAPE values simulated are up to 1600 J kg$^{-1}$ (Fig. 15c).

    At 07:00 UTC an increasing number of air parcels found inside updrafts over the Aude department originate from south of the Balearic Islands, i.e. directly from CF2, the front formed with Leslie's remnants (Fig. 14a) in comparison with the first part. These parcels, projected in Fig. 14c originate from altitudes between 1 and 4 km, generally higher than what was simulated in the first part. Some air parcels carried by the LLJ, coming from the east of the domain and altitudes between 0 and 1 km

are still found and projected in Fig. 14b. If their number decreased compared to the first part, their water vapour mixing ratio remains high with 10.4 g kg$^{-1}$ on average, which increased through their transport above the Mediterranean Sea by 1.8 g kg$^{-1}$ (Tab. 1). Some descending dry air parcels are also found (Fig. 14d), originating from dry mid-level areas located at the rear of CF2.

    In the lower levels, at 697 m, REF simulates a strong wind variation along CF2 (Fig. 15b): ahead of CF2, the southeasterly

LLJ reaches 22 to 26 m s$^{-1}$ while at the rear wind turns southwesterly and only reaches 6 to 12 m s$^{-1}$. Between 07:00 UTC and 09:30 UTC, the northeastwards advance of CF2 propagated this wind variation over the Aude department. When wind speed decreased along CF1, the cold air west of CF1 started flowing eastwards, towards the Mediterranean Sea, as a density current, spreading out circularly over the sea. This cold air rapidly flowing as soon as the LLJ stops tends to show that an equilibrium maintaining CF1 quasi-stationary was reached between CF1 and the LLJ. Then, the equilibrium progressively broke from south

to north by the advance of CF2.

**Figure 14.** As Fig. 11 for trajectories ending at 07:00 UTC 15 October at 43.26° N, 2.34° E. Air parcels are projected along the same axis A–B but separated depending on their origin and behaviour: (b) ascending parcels from east, (c) ascending parcels from south or west and (d) descending parcels from south.



**Figure 15.** As Fig. 12 at 07:00 UTC 15 October.

## 5.3 Amount of moisture supplied by Leslie's remnants

To quantify the amount of moisture brought over the Aude department during the HPE by Leslie's remnants, the geographic origin of air parcels found in the atmospheric column above the 338 mm precipitation maximum simulated by REF (see Fig. 6b, now referred to as $C_{338}$) is tracked. Every 30 min from 19:00 UTC 14 October to 10:00 UTC 15 October (the time period of the HPE in the REF simulation), backward trajectories of 36 air parcels taken every 2 vertical levels of the model between 18 m and 9924 m height are computed until 12:00 UTC 14 October, the beginning of the REF simulation. Here, parcels above 10 km height are not taken because most parcels followed above this height are not found inside any updraft over the Aude department.

The computation of backward trajectories provides the initial parcel location: it is the location of the parcel at 12:00 UTC 14 October or if the parcel is out of the parent domain at that time, it is the location where the parcel enters the parent domain. The





**Table 2.** Properties and geographic origin of air parcels found in the atmospheric column $C_{338}$ as a function of time, aggregated in time intervals from 19:00 UTC 14 October to 10:00 UTC 15 October. Mean water vapour mixing ratio 2 h before reaching $C_{338}$ (i.e. before some parcels are lifted) is indicated by $\bar{r}_v$.

| Initial parcel location | Properties | Time interval (hours in UTC) | | | | | | | | |
|---|---|---|---|---|---|---|---|---|---|---|
| | | [19–21[ | [21–23[ | [23–01[ | [01–03[ | [03–05[ | [05–07[ | [07–09[ | [09–10[ | Total (%) |
| East of 4.5° E | Number | **76** | **67** | **76** | **70** | **75** | **52** | 25 | 30 | **471 (42 %)** |
| | $\bar{r}_v$ (g kg$^{-1}$) | 10.6 | 9.7 | 9.4 | 9.4 | 8.7 | 9.0 | 10.2 | 9.7 | **9.5 (54 %)** |
| West of 4.5° E – south of 42° N | Number | 66 | 55 | 48 | 33 | 31 | 39 | **64** | **44** | 380 (34 %) |
| | $\bar{r}_v$ (g kg$^{-1}$) | 2.6 | 1.7 | 3.7 | 4.6 | 4.8 | 4.9 | 4.7 | 4.1 | 3.7 (17 %) |
| Other | Number | 2 | 22 | 20 | 41 | 38 | 53 | 55 | 34 | 265 (24 %) |
| | $\bar{r}_v$ (g kg$^{-1}$) | 9.6 | 9.9 | 10.2 | 9.6 | 9.3 | 8.9 | 8.9 | 8.4 | 9.2 (29 %) |

geographical origin is divided in 3 categories (Tab. 2): east of 4.5° E, west of 4.5° E – south of 42° N and other. Such division is done because air parcels from Leslie's remnants originate from the area west of 4.5° E – south of 42° N. The "other" category includes mostly cold and stable air parcels. These parcels are found at the lowest levels of the atmospheric column because it is located west of CF1 (such parcels are shown in Fig. 11b). Because these parcels are stable, only the first categories including

conditionally unstable air parcels are compared.

    Tab. 2 shows that 42 % of air parcels found in $C_{338}$ originated from east of 4.5° E, i.e. from east of the Balearic Island, and they carried 54 % of the water vapour mixing ratio of all air parcels tracked. On the contrary, 34 % of air parcels coming from Leslie's remnants carried only 17 % of the water vapour mixing ratio, although their number particularly increased in the second part of the event. This result suggests that air parcels coming from east of 4.5° E, over the Mediterranean Sea, supplied

more moisture to convective cells than air parcels originating from Leslie's remnants.

### 5.4 Lifting by the Albera Massif and effect on precipitation

To understand the role of the Albera Massif in generating convective bands downwind, as it is observed and simulated by REF, the simulations REF and NOALB are compared.

    Heavy precipitation is simulated at 05:00 UTC over the highest slopes of the Albera Massif and along a line downstream of

the massif (Fig. 16a, dashed red area). When the Albera Massif is removed, Fig. 16b shows that no precipitation is simulated over or downstream of it. In NOALB, along the eastern Pyrenees, west of the dashed red line, a slightly larger area of instantaneous precipitation above 10 mm h$^{-1}$ is found than in REF. Fig. 17a shows that over and downstream of the Albera Massif, heavy precipitation in REF results from aligned convective cells exhibiting ascending vertical velocities above 4 m s$^{-1}$, whereas no substantial ascending movement is found along this line in NOALB (Fig. 17b). The orientation of the line of updrafts (along

the A–B axis in Fig. 17) is parallel to the horizontal wind streamlines at 2674 m height, showing that convective cells are aligned with the south-southeastern mid-level wind direction. Superimposed with this line of updrafts, lee waves resulting



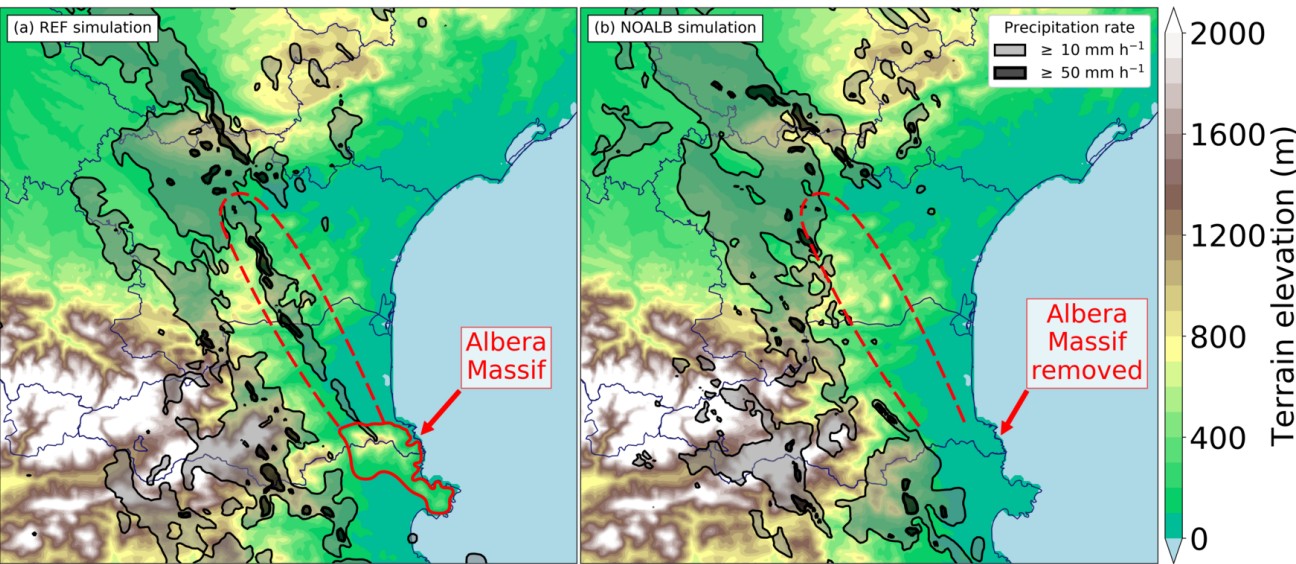

**Figure 16.** Instantaneous precipitation rate at 05:00 UTC 15 October superimposed on the terrain elevation of (a) REF and (b) NOALB simulations.

in quasi-stationary, evenly spaced, couples of positive and negative vertical velocities indicated by the large black arrows in Fig. 17a are simulated by REF but are not found in NOALB (Fig. 17b).

To quantify the flow regime of the situation at 05:00 UTC 15 October, the mountain Froude number $Fr_m = \frac{U}{Nh}$ (Kirshbaum

et al., 2018) is estimated, where U is the mean wind speed of the layer, N is the Brunt-Väisälä frequency and h is the mountain height. Here h = 1128 m, the maximum height of the Albera Massif in the model. To compute N, the bulk method described by Reinecke and Durran (2008) is used considering a single layer which has approximately the height of the mountain: N = $\sqrt{\frac{g}{\theta}\frac{\theta_{(25)}-\theta_{(1)}}{h}}$, where g = 9.81 m s$^{-2}$ is the standard acceleration of gravity, $\theta_{(n)}$ is the potential temperature at model level n (first level is at 5 m height and the 25$^{th}$ is at 1143 m height) and $\bar{\theta}$ is the mean potential temperature over the layer. The moist

Froude number $Fr_w = \frac{U}{N_w h}$ (Chen and Lin, 2005) is also computed, where $N_w$ is the moist Brunt-Väisälä frequency that differs from N because $\theta$ is replaced by $\theta_v$. The computation of U, N and $N_w$ is an average over 100 grid points located upwind the mountain, precisely the grid points less than 10 km east and 10 km south of grid point A (A is shown in Fig. 17a). Computation leads to U = 21.3 m s$^{-1}$, N = 9.9 × 10$^{-3}$ s$^{-1}$, $N_w$ = 8.9 × 10$^{-3}$ s$^{-1}$ which gives $Fr_m$ = 1.9 and $Fr_w$ = 2.1. According to Kirshbaum et al. (2018), $Fr_m$ = 1.9 > 1 indicates that the flow tends to directly ascend the terrain over the windward slope instead of being

deflected around the obstacle. This ascent mechanically lifts the conditionally unstable air parcels supplied by the LLJ, but also probably triggers the aforementioned lee waves. $Fr_w$ = 2.1 corresponds to the flow regime IV of Chen and Lin (2005), described as a flow with an orographic stratiform (a stratiform cloud is defined by the authors as having a cloud depth less than 4 km) precipitation system over the mountain and possibly a downstream-propagating cloud system.





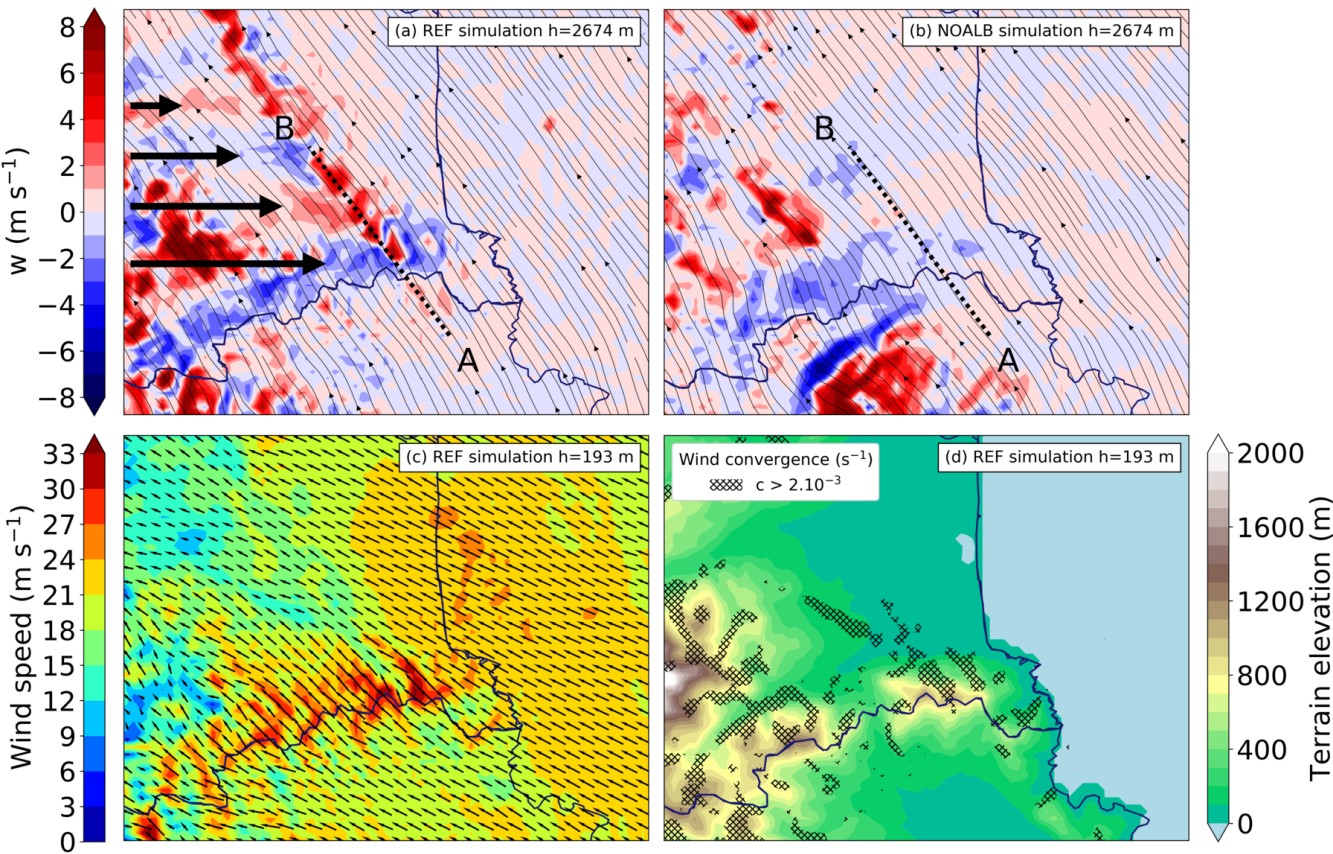

**Figure 17.** Vertical velocity (w) and horizontal wind streamlines at 2674 m height in the (a) REF and (b) NOALB simulations. Horizontal wind (c) speed and direction and (d) convergence $> 2 \times 10^{-3}$ s$^{-1}$ at 193 m height in the REF simulation at 05:00 UTC 15 October.

To closely look at how convective cells are initiated and maintained, a time evolution of the vertical cross section A–B

simulated by REF is shown in Fig. 18. At 05:05 UTC, a convective cell containing hydrometeors is formed above the Albera Massif (Fig. 18a, black arrow). Inside this cell, the potential temperature is higher than the environment around 2 km height probably due to latent heat release associated with water phase changes. This cell is advected towards B by the mid-level wind. A second cell is initiated at the rear of the first one by orographic lifting and the surface of hydrometeor mixing ratio above 1 g kg$^{-1}$ has rapidly increased (Fig. 18b, brown arrow). On the lee side of the mountain, these convective cells propagate in

the middle of a large subsidence area (related to the aforementioned lee wave), but it seems that counteracting the effect of the subsidence, a quasi-stationary wind convergence zone located near the ground connects with the updrafts and invigorates them. This wind convergence zone is simulated downwind of the mountain (Fig. 17d) and seems due to rapid wind decrease and some wind confluence in the lee side of the Albera Massif (Fig. 17c). Then, as shown by Figs. 18d,e, both cells connect with an ascending zone of the lee wave around 20 km along the axis A–B and rapidly grow: the hydrometeor mixing ratio surface

above 1 g kg$^{-1}$ as well as the vertical velocity are found to rapidly increase inside both cells. These findings are consistent



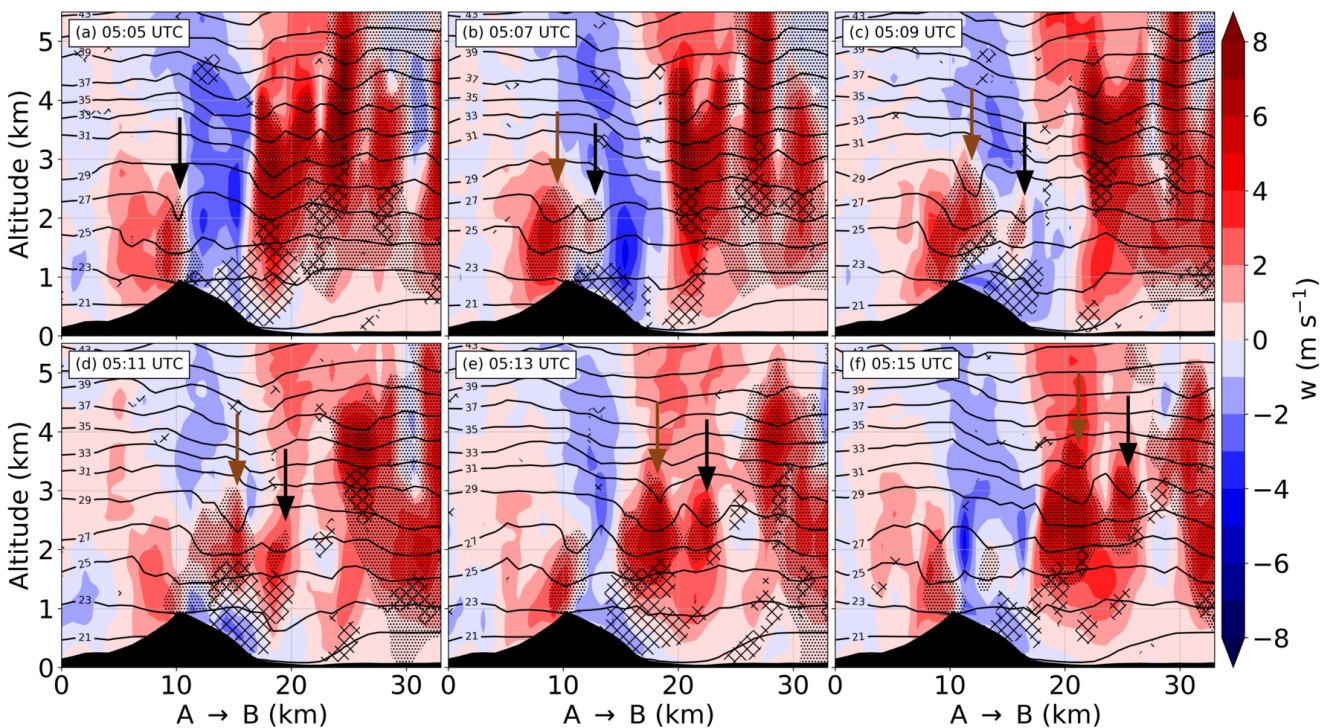

**Figure 18.** Vertical velocity (w, m s$^{-1}$), potential temperature (black contours, °C), horizontal convergence $> 2 \times 10^{-3}$ s$^{-1}$ (black diagonal hatches) and hydrometeor mixing ratio $> 1$ g kg$^{-1}$ (little black dots) between 05:05 and 05:15 UTC 15 October. Terrain is in black.

with the description of the flow regime IV of Chen and Lin (2005): vertical velocities above 1 m s$^{-1}$ and hydrometeor mixing ratios above 1 g kg$^{-1}$ remain below 4 km altitude over the mountain, and convective cells are propagating downstream of the mountain.

Because of the slightly directional vertical wind shear simulated in the lower part of the troposphere (see wind direction
in Figs. 17a,c), the LLJ continuously supplies conditionally unstable air parcels to the convective cells formed from their southeastern flank while these cells are advected by the south-southeastern mid-level wind. Backward trajectories starting from their updrafts (not shown) show that once convective cells are on the lee side of the mountain, as they are advected north-northwestwards, an increasing number of low-level moist air parcels that have not crossed the Albera Massif are found inside the cells. Thus, the supply of conditionally unstable air parcels along the line is continuous and possibly explains the
maintenance of the convective cells long after they are formed.

This preferential organization of convection along a line downstream of the Albera Massif results in substantial departures in 24 h rainfall accumulations (Figs. 6b,d) between NOALB and REF. Precipitation along the band downstream of the Albera Massif called eastern band in Sect. 2 is reduced by as much as 100 mm in NOALB compared to REF. REF maximum precipitation over plains is reduced from 338 mm to 310 mm, and the maximum in NOALB (332 mm) is shifted over mountains.





Concomitantly, precipitation is enhanced along the western precipitation band (see Sect. 2) downstream of the eastern slopes
of the Pyrenees: maximum precipitation is increased from 296 mm to 327 mm in NOALB compared to REF. Precipitation is
also enhanced between both bands, along the quasi-stationary CF1. Consequently, the southeast-northwest orientation of the
precipitation zone exceeding 240 mm in REF is replaced by a south-north orientation in NOALB.

This sensitivity experiment shows the large importance of the Albera Massif in the shape of the precipitation field, focusing
precipitation downstream of it while reducing precipitation elsewhere. The proposed mechanism describing convection initia-
tion over the Albera Massif, one of the first relief intercepting the marine LLJ and the convective cells maintenance downstream
of these may be applied to similar reliefs. It may explain the enhanced amount of precipitation observed along the western band
and the formation of the third line starting over the Corbières Massif observed between 00:00 and 02:00 UTC (Fig. 5c).

## 6    Influence of the cooling associated with the evaporation of precipitation

This section investigates the possible influence of the cooling associated with the evaporation of precipitation over the Aude
department on CF1. The following questions are addressed. Does this process (i) modify the location of CF1? (ii) extend the
duration of CF1 stationarity? (iii) enhance the temperature gradient along CF1?

Fig. 6c shows the resulting 24 h accumulated precipitation observed in the NOCOOL simulation. Precipitation is globally
higher in NOCOOL than in REF, maximum precipitation reaches 415 mm in NOCOOL and only 338 mm in REF. Compared
to REF, a local maximum of precipitation of 310 mm located north-northwest of the global maximum appears in NOCOOL.
The organization of the precipitation in two major bands remains but their orientation is slightly rotated clockwise compared
to REF.

At 04:00 UTC, both REF and NOCOOL exhibit a sharp east-west horizontal $\theta_v$ gradient that delineates the location of CF1
(Fig. 19). At 5 m height, REF shows about 0.5 to 1 °C colder temperatures than NOCOOL on both sides of CF1. Highest
departures are found over the Pyrenees. South of the A–B axis, CF1 is shifted from 0 to 10 km west in NOCOOL compared
to REF, depending on the latitude. Along the A–B axis, vertical cross sections of Figs. 19c,d show that CF1 is about 90 km
from A in NOCOOL and 87 km from A in REF, near the ground. The $\theta_v$ gradient is visible up to 2 km in both simulations.
East of CF1, near the ground, virtual potential temperatures of about 22 °C are found in both simulations. West of CF1, these
temperatures are only found at altitudes between 1000 and 1500 m in REF and between 875 and 1400 m in NOCOOL. West of
CF1, inside the cold air, the altitude of the isotherm $\theta_v = 18$ °C is higher in REF with about 450 m than in NOCOOL with about
325 m. Above 750 m a.s.l., the vertical temperature gradient is stronger in NOCOOL than in REF, thus REF shows slightly
lower temperatures than NOCOOL from 750 to 2500 m.

Along the A–B axis, departures of generally less than 0.5 °C are found between simulations east of CF1. West of CF1,
generally higher temperatures are found in NOCOOL compared to REF, with temperatures up to 2 °C higher found in NO-
COOL above 1500 m and near B. Near CF1 and near the ground, departures remain generally below 1 °C. The evaporative
cooling does not shift the location of CF1 by more than few kilometres or substantially modify the temperature gradient along
it. Comparison of REF and NOCOOL does not show any extended duration of CF1 stationarity or substantial time lag between



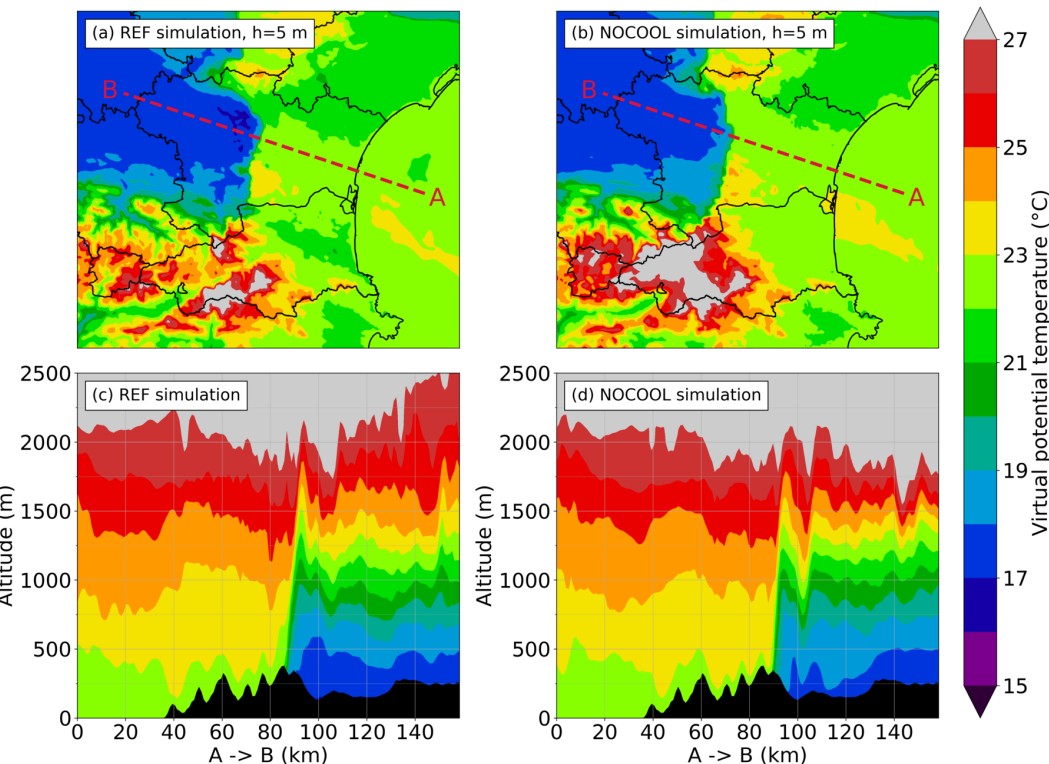

**Figure 19.** Virtual potential temperature at 5 m height for the (a) REF and (b) NOCOOL simulations at 04:00 UTC 15 October. Vertical cross sections of virtual potential temperature along the A–B axis are drawn for the (c) REF and (d) NOCOOL simulations.

simulations. However, globally higher temperatures are simulated in NOCOOL than in REF in the lowest troposphere because evaporative cooling is switched off. These higher temperatures result in globally higher MUCAPE (not shown) and conse-

quently stronger convective cells and stronger rain rates in NOCOOL than in REF. This probably explains why precipitation is substantially higher in NOCOOL than in REF.

One of the reasons that may have limited the evaporative cooling west of CF1, where highest precipitation is observed, is the small evaporation due to near saturation of air masses in the lower troposphere according to the REF simulation (Fig. 20). At 04:00 UTC, relative humidity exceeded 90 % over most of Languedoc-Roussillon and a dry air mass was only found over

Spain, at the rear of CF2. Large evaporative cooling associated with dry levels between 1000 and 4000 m is not simulated in this case. Such difference may explain why the location of CF1 between REF and NOCOOL is similar. The cooling associated with evaporation processes does not have any substantial impact on the stationarity of the simulated MCS on the 2018 Aude HPE, agreeing with the same observation of Ducrocq et al. (2008) on the 1999 Aude HPE.

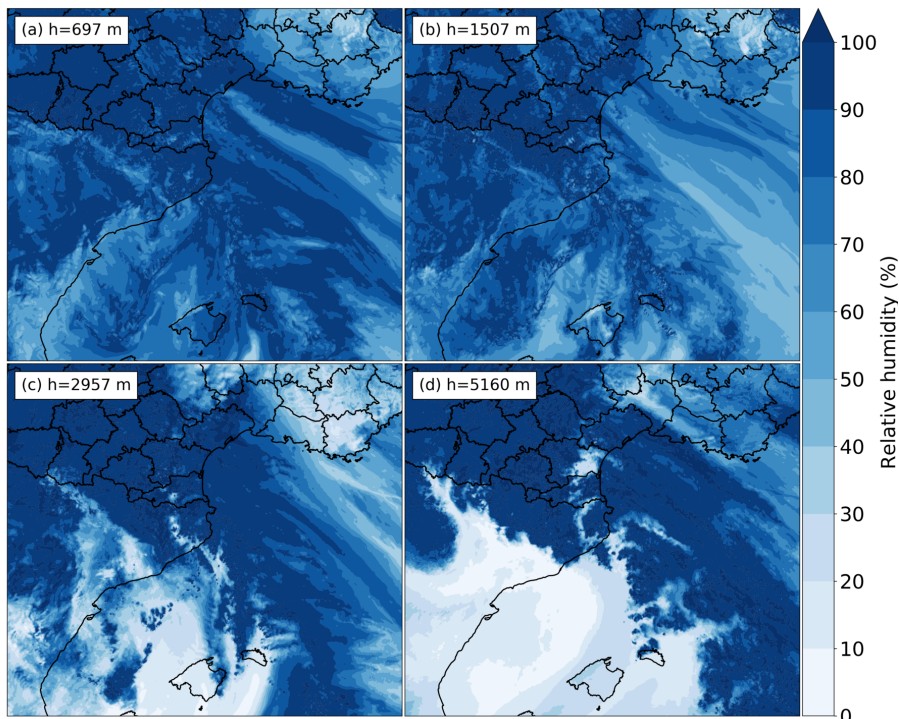

**Figure 20.** Relative humidity in the REF simulation at 04:00 UTC 15 October at (a) 697 m height (b) 1507 m height (c) 2957 m height (d) 5160 m height.

## 7 Conclusions

The meteorological situation of the 14 and 15 October 2018 exhibited favourable ingredients leading to a HPE over Languedoc-Roussillon and particularly the Aude department. The remnants of the former hurricane Leslie are involved in the formation of a Mediterranean surface low and its associated cold front (CF2). The rapid deepening of this Mediterranean surface low, extended by a trough over Languedoc-Roussillon, contributed to strengthen a low-level jet (LLJ) over the Mediterranean Sea. Meanwhile, a cold front (CF1) remained quasi-stationary in the middle of the Aude department, west of the trough. The slow 545 movement northwards of the Mediterranean low as well as the quasi-stationarity of its associated trough sustained quasi-stationary atmospheric conditions during several hours over the Aude department, while continuously supplying conditionally unstable air parcels over the area.

A two-way grid nested numerical simulation at 1 km and 500 m horizontal resolutions was successfully carried out with the Meso-NH model with initial and boundary conditions provided by the operational AROME analyses every 3 h. A comparison 550 of the simulation with near-surface analyses built from observations of standard and personal weather stations was performed. Compared to observations, the simulation delays by approximately 3 h the arrival of the precipitation system ahead of CF2 over the Aude department as well as the end of the HPE over the west of this department. The location where both CF1 and the




trough remained quasi-stationary is well simulated, around 10 km farther west than in analyses, but the stationarity duration of both low-level mesoscale boundaries is overestimated by 3 to 4.5 h. Simulated 24 h accumulated precipitation is found

realistic but slightly overestimated compared to observations. This overestimation is consistent with the delay that caused longer precipitation over the Aude department. Despite these differences, the simulation is considered realistic enough and taken as the reference (REF) to study the case.

The study reveals that the main origin of lifted air parcels and dominant mechanisms that triggered convection differ during the two parts of this HPE.

The first part begins around 19:00 UTC 14 October in both observations and the REF simulation. In this part, the REF simulation shows that conditionally unstable air parcels originating from the Mediterranean Sea, east of the Balearic Islands at altitudes between 0 and 1.5 km are carried by the LLJ towards the Languedoc-Roussillon shore. During their transport above the Mediterranean Sea, inside the model's domain, the water vapour mixing ratio of the tracked air parcels increases by 0.5 to 1.8 g kg$^{-1}$. Once inland, these air parcels are continuously lifted over the upwind slopes of the first mountains encountered,

i.e. mostly eastern Pyrenees relief, the Albera Massif and the Corbières Massif. Both observation and simulation show that convective cells organize along quasi-stationary lines downwind of the mountains, forming the active parts of a back-building MCS. Convective cells appear to be maintained and reinforced downwind of the terrain by low-level leeward convergence, ascending areas created by mountain lee waves and favoured supply in conditionally unstable air due to the low-level directional wind shear. Indeed, formed cells are continuously advected north-northwestwards by the mid-level wind while the LLJ supplies

conditionally unstable air on their southeastern side. A sensitivity study shows that the convective line downwind of the Albera Massif disappears when the Albera Massif terrain is flattened, showing the crucial role of terrain in the formation of these lines. Convection is particularly enhanced above and west of the quasi-stationary cold front CF1, along which a strong wind convergence line as well as a substantial virtual potential temperature gradient are simulated and observed. Most parcels found in large updrafts near the simulated precipitation maximum, west of CF1, are found to be lifted above CF1.

The second part begins around 02:00 UTC 15 October in observations and 05:00 UTC in the REF simulation. In this part, the REF simulation shows that an increasing number of conditionally unstable air parcels originates from south of the Balearic Islands i.e. from the vicinity of the Mediterranean low and CF2, both formed with Leslie's remnants, at altitudes mostly between 1 and 4 km. The advance of CF2 and the conditionally unstable air mass located ahead of it trigger convective cells over the Mediterranean Sea that are advected towards the Languedoc-Roussillon coast. Inland, the MCS loses progressively its

organization in lines in this second part and the strongest rain rates are found along and west of CF1. The end of the HPE over the Aude department is driven by the advance of CF2 northeastwards.

Regarding the quasi-stationary location of CF1, the REF simulation indicates that CF1 reached a sort of equilibrium with the LLJ blowing in the opposite direction. This equilibrium broke when the wind speed dropped, which caused the cold air mass west of CF1 to flow rapidly eastwards. A sensitivity study shows that evaporative cooling plays no role in the stationarity of

CF1. One reason may be the low evaporation due to the near saturation of the middle and lower troposphere, to which Leslie's remnants contributed. The location where CF1 remained quasi-stationary appears correlated with the location of the maximum precipitation in both simulation and observation.



Consequently, in order of importance, the location of the exceptional precipitation over the Aude department seems the result of the convective activity focusing (i) west of the quasi-stationary CF1 and (ii) downwind of the Albera Massif and the
Corbières Massif. Precipitation maximum is found at the junction between these areas in both simulation and observation. Regarding the role of Leslie's remnants, they (i) are involved in the formation of the cold front CF2 behind which a Mediterranean low deepened rapidly, (ii) contributed to the supply of low-level conditionally unstable air in the second part of the event and (iii) contributed to moisten mid-levels of the troposphere, diminishing evaporation processes. However, low-level moisture that contributed to precipitation over the Aude department mainly originated from the Mediterranean Sea, east of $4.5° E$, rather
than Leslie's remnants.

Future work could quantify Leslie's contribution in the cyclogenesis mechanisms of this Mediterranean low and more generally the role of ex-tropical cyclones in disturbing weather of the Mediterranean basin. Accurately track and represent the life cycle of Mediterranean lows seems crucial to better forecast HPEs. The use of near-surface analyses built from a large number of standard and personal weather stations, independent of the model studied, was beneficial in this case, helping to
understand the inevitable departures between simulation and observation. It also highlighted small-scale movements of the quasi-stationary front CF1 that are simulated by the model but cannot be seized by the surface network. Future work could focus on real time tracking of low-level mesoscale boundaries which are involved in the stationarity of precipitation. In order to do that, rapidly updated near-surface thermodynamical analyses, only derived from observations and independent of models, seems promising. They could allow forecasters to evaluate in real time the accuracy of near-surface fields forecasted by
numerical models. In parallel, the assimilation of personal weather stations, supplementary observations that contributed to the quality of the near-surface analyses used in this study, could be tested in convective-scale models like AROME.

*Code availability.* The Meso-NH model is freely available online at http://mesonh.aero.obs-mip.fr/mesonh54.

*Data availability.* Standard weather stations data, manual rain gauges data, radar mosaic, ANTILOPE, ARPEGE and AROME analyses are provided by and property of Météo-France. World Meteorological Organization essential weather stations data, radar mosaic at a 15 min
time step, ARPEGE and AROME models are available in real time at https://donneespubliques.meteofrance.fr/. Direct online access of most datasets used in the study is not available: most of them can be provided on demand by the corresponding author only for research purposes.

*Author contributions.* This work was carried out by MM as part of his PhD thesis under the supervision of OC. MM and OC designed the study, interpreted the results and wrote the paper.

*Competing interests.* The authors declare that they have no conflict of interest.



*Acknowledgements.*   This work is a contribution to the HyMeX programme supported by MISTRALS. We wish to particularly thank Quentin
Rodier as well as the PHY-NH and PRECIP teams for valuable discussions about the Meso-NH model.



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
