# Peer review of "A numerical study to investigate the roles of former hurricane Leslie, orography, and evaporative cooling in the 2018 Aude heavy precipitation event"

_Weather and Climate Dynamics, 2020_

## Referee Comment (RC1) · Anonymous Referee #1 · 1 Dec 2020

Generic comments

This study investigates the mesoscale dynamics of the heavy precipitation event affecting the Aude region during the heavy precipitation event of the 14-15 October 2018. After a (too ?) long and detailed description, the roles of former hurricane Leslie, the orography, and evaporative cooling are examined. The interaction of the incoming trough and surface cyclone with the orography helped the convective activity focusing west of the quasi-stationary cold front and downwind of the Albera Massif. Leslie's remnants are involved in the formation of the second cold front CF2 and contributed

to the supply of low-level conditionally unstable air in the second part of the event. However, the grater contribution to the precipitation over the Aude department mainly originated from moisture coming from the Mediterranean Sea. Finally evaporative cooling did not seem to play a substantial role in the dynamics but only in the control of the total simulated amount of precip.

The paper is well written and clearly structured however I feel it sometimes goes too long in the description of the dynamics of the event in a disproportionate way compared to the focus of the research. The analysis of Leslie's contribution and evaporative cooling is resolved in a short part compared to the long introduction on the analysis of the event and the methodology of analysis which however I find accurate.

So I do not have a strong recommendation if not to make the article more concise and to the point.

Specific comments

I'm doubtful on the expression "personal weather station". What about "private weather stations" belonging to citizen weather observing networks integrating the official MeteoFrance Network. Or something similar. I would also put a reference if this initiative is coordinated by the National Met Service.

The literature cited in the introduction is appropriate. I think that the synoptic description, as well the numerical simulations section can be shortened perhaps limiting to the essential level of details needed for the following dynamical assessment.

I'm also suggesting to shorten the Conclusions eliminating unnecessary details like time of the day references of values of single variables to have a more compact way to present results. In the conclusions bullet points (line 590), I would add the part on the evaporative cooling reported earlier at line 585

---

## Referee Comment (RC2) · Anonymous Referee #2 · 7 Jan 2021

Interactive comment on "A numerical study to investigate the roles of former hurricane Leslie, orography, and evaporative cooling in the 2018Aude heavy precipitation event" By Marc Mandement and Olivier Caumont Anonymous Referee #2

Generic comments This study focuses on the investigation of a heavy precipitation event in October 2018 in the Aude region, south-eastern France. Following the investigation by Caumont et al. (2020), there main aspects are evaluated in this investigation, the origin of moisture with particular attention paid to the role of the hurricane Leslie, the role of the Pyrenees and the role of the evaporative cooling. Numerical simulations

and near-surface observations are examined with this purpose. Given the exceptionality of the event, but the increase in the frequency of such situations, and the enormous fatal consequences, the understanding of this heavy precipitation event is of relevance. The paper is clear and well structured. However, sometimes too descriptive, which could make it difficult for the reader to follow the main investigation line of the article. Therefore, I suggest shortening too descriptive sections and focus with more detail in the description of those aspects pointed out as goals of this investigation. The literature cited in the introduction is appropriate although I suggest additionally introducing more recent references, also providing some information and references of previous publications, focusing on the key role of moisture and its origin, topography, and evaporative cooling on heavy precipitation events.

Specific comments 1. It could be important to clearly state in the abstract and more in detail in the introduction what is the novelty and contribution of this investigation. 2. The model and resolution used for the numerical simulations could be provided already in the abstract. 3. Try to substitute 4.5° in the abstract for a "easier location definition" for the reader. 4. The autumn 2018, and particularly October, was a period in which the western Mediterranean region was strongly affected by damaging heavy precipitation events, including the known as the Aude 2018 event. Also, on 9 October 2018 the Balearic Islands suffered catastrophic consequences due to similar heavy precipitation phenomena. Thus, the manuscript will benefit from a description of the conditions in the autumn period for the western Mediterranean, the comparison with the climatological conditions particularly for the north-western Mediterranean, and the description of any possible connection between the two events. 5. The case description and the numerical simulations section could be shortened with major focus on the information relevant for the case analysis and answering of the questions initially raised. 6. Please rethink the sentence in L215 "... is run to realistically simulate the case". 7. It would be interesting providing an additional simulation in which the Pyrenees were removed since its position is upstream and one goal of this study is provide and analysis of the role of topography. If this is not possible, a

hypothesis would be needed. 8. Please provide information of why only near-surface information is analysed and validated. 9. There are too many figures in the article, and multiple panels in some of them. Please, restrict the number of figures to those strictly necessary and try to combine when possible several information in one figure.

Please also note the supplement to this comment:
https://wcd.copernicus.org/preprints/wcd-2020-54/wcd-2020-54-RC2-supplement.pdf

---

## Referee Comment (RC3) · Anonymous Referee #3 · 4 Feb 2021

**General comments**

- Paper would benefit from a more diverse collection of references, which place these heavy rainfall events into wider context and better draw upon literature from all across the world.
- A schematic diagram of typical synoptic setup during heavy precipitation event might be useful (you show a map in Figure 1, but as the reader, I want more information and detail). This kind of plot would really sell the paper to the reader.
- The detail in evaluating model performance against observations is impressive.
- The length of this paper could be reduced. The level of detail is commendable, but there are sections that are difficult to fully take in because of the consistent level of depth.
    - The text on the model setup in Section 3.1 is an example. The level of detail is impressive, but do you need to go into this much detail?
    - Along the same line, you could lose a couple of figures (and accompanying text) without reducing the quality of your writing. As an example, you could remove either Figure 7 or 8, Figure 10, and Figure 13 without impacting the quality of your analysis.
- Careful to use the same tense throughout your writing. In places you switch between present and past, which is confusing.
- When you introduce diagnostics such as water vapour mixing ratio and precipitable water in Section 2, it would be useful to know how large these values are relative to climatology (how unusual?).
- Splitting section 2 into two sub-sections, one focusing on the synoptic evolution and the other on the mesoscale details of the rainfall, could help to make the writing more streamlined and easier to follow. At the moment, there is too much information crammed in; the section is too long.
- You have confused 'westwards' and 'eastwards' in places throughout the text. Make sure that your descriptions are accurate and consistent.
- There are quite a few occasions where you describe details of the synoptic or mesoscale evolution without referring to figures. It is fine to do this on a few occasions (adding 'not shown'), but not too much, or the reader will get confused.
- Some of the more technical information could be included in an Appendices section at the end of the paper, rather than in the main text (would help to streamline the text). The first paragraph in Section 4 is a good example (on REF_SP).
- Generally, there is too much description of results, and not enough interpretation and putting your results into wider context.
- When explaining to the reader why this study is important (Introduction), you should discuss the topic of interactions between tropical cyclones and the midlatitude flow, and how they can impact upon predictability. Papers by

Christian Grams and Florian Pantillon provide good examples of this type of work. This type of discussion will help to link the specifics of this event that you're discussing with larger-scale issues of interest in the meteorology community.

**Specific comments**

- Avoid vague language (a few replacements are suggested)
    - convective activity → convection
    - synoptic situation
    - seem to be maintained → maintained
    - density departures
    - thermal signature
    - more reflectivities are observed
    - precipitating activity → precipitation
- Standard and personal weather stations – please elaborate
- You introduce the concept of the stationarity of the cold front (L11) without any prior discussion – seems a bit rushed
- L15 to 17: 'the location of the exceptional precipitation appears to be driven primarily by the location of the quasi-stationary cold front…' seems like an obvious statement to make. Would like more insight here.
- Which dataset did Ricard et al. (2012) use for their climatology of heavy precipitation events over the northwest Mediterranean?
- 'Mediterranean Sea supplied up to 60% of the total air parcels moisture (Duffourg and Ducrocq, 2013; Duffourg et al. 2018), modulating the intensity of convective precipitation.' How did the authors calculate this value (60%)? More information on the method is needed here, even if brief.
- L46: don't start a new sentence with 'this'. Always refer directly to the part of the previous sentence that you're referencing (i.e. 'This combination of factors…').
- L52: need a sentence to tell the reader that you're introducing your case study here. Currently, you just start talking and the transition from the topic in the previous paragraph is not smooth enough.
- Paragraph on the event itself reads well (L52-62); you summarise the key points nicely.
- L68-69: 'Because of the heavy rain observed in the area, evaporative cooling processes may have played a role in the stationarity of the cold front.' Not sure I follow this argument. Evaporative cooling would be expected in conjunction with the rainfall, but why are you hypothesising that this cooling could play an important role in the movement of the front? You need to make this point more confidently here, and with more detail.
- L69-71: link between evaporative cooling and conditionally unstable air? Are you hypothesising that evaporative cooling occurred in the mid troposphere and destabilised the profile? Need to make these details clearer.

- L84-87: where is the evidence that the interaction between both lows seems to have strongly slowed the westward movement of the mid-level cut-off low? Or, do you mean eastward movement? This would make more sense.
- L88-92: which figure are you referring to in this discussion? Make the connection clearer and add detail to figure if necessary.
- L103-104: "A potential vorticity at upper levels is observed upstream of the low." Where is the evidence for this feature? Need to relate all statements to figures, or add '(not shown)'.
- L104: "It may have helped to deepen it…". A reference or two here on the interaction between surface cyclones and upper-level PV anomalies would be beneficial. Something like Hoskins et al. (1985), or a classic paper along those lines.
- Be as precise as possible in your discussion of figures. For example, in Figure 4 you plot the 925 hPa water vapour mixing ratio and geopotential height. In the accompanying text, make sure that you refer to the diagnostics in the figure ("925 hPa wind"), rather than using more vague descriptions ("near-surface wind").
- Do the authors have any theory on why precipitation within the western band is overestimated by the model, or why the model overestimates precipitation over orography?
- Add latitude and longitude labels and tickmarks to Figures 2 to 6
- Overlay the position of key features such as CF1 and CF2 on the relevant figures. It would make the connection between text and figures much stronger.
- How do Leslie's remnants contribute to the formation of the cold front CF2, behind which a Mediterranean low deepened rapidly?
- Mark the Albera Massif on Figure 5. Also mark the Aude and Pyrénées-Orientales departments. You refer to these features, but the reader won't necessarily know where they are (without referring back to earlier figures).
- The convective lines that you mark on Figure 5 are not easy to see. The eastern line in Figs. 5a to c is fine, but the other two lines are much more difficult to pick out. Is there another way that you could annotate these figure panels?
- L229: do you mean downstream? Upstream suggests the windward (not lee) side of the ridges.
- L243-255: when justifying your investigation of evaporative cooling, it's not immediately clear how evaporative cooling near the surface could modify the location of CF1 or change its stationarity. Could you describe how this could occur (maybe add a reference)?
- L248-255: in the NOCOOL simulation, did you turn evaporation off completely in the child domain, or just set the temperature tendency from evaporation to zero? Assuming the second – it wouldn't make a difference to your results, but it's best to be as precise as possible when discussing the changes you made to the model output.

- L277-286: you discuss the evolution of precipitation in your model simulation relative to observations, but don't show any of the plots required to do so (Figure 6 presents accumulated rainfall but gives no indication of how the structure of the rainfall evolves during the 24-h period).
- Figure 8: why have you used virtual potential temperature rather than potential temperature to diagnose the position of the front? Would be interesting to know what the pure temperature difference across the front looks like (potential temperature) as well as the moisture difference (virtual potential temperature). Don't change anything in the manuscript, it's just something to think about in future.
- Figure 10: the idea is good, but it's difficult to pick out the difference between the observations and the REF simulation, because as the reader you are drawn to the filled contours (REF winds) and it's hard to see the differences between the wind vectors in black and red. Given that the text accompanying this figure makes a similar point to that with Figure 9, I would recommend removing the figure. You could still make your point about the longer stationary period in the REF simulation (relative to observations) likely explaining some of the over-estimation in accumulated rainfall.
- L345-346: how does the Lagrangian trajectory tool of Gheusi and Stein (2002) differ from newer tools such as LAGRANTO?
- Discussion of Figure 11 (~L355): make it clearer in the figure panels and in the text that the trajectories in (b) correspond to the transect (A → D), etc. The reader will get confused otherwise. You have done this in your discussion of Figure 13 → please apply the same method to the discussion of Figure 11.
- Nice illustration of lifting along the cold front in Fig. 11c.
- Overlay the position of the cold fronts (CF1 and CF2) in Figures 11 and 12, and on Figures 14 and 15.
- L387-389: the claim that the trough over the Aude department may locally increase wind shear inland, based on Fig. 10a, is not based up strongly by the evidence.
- Do you need Figure 13? The results are similar to those in Figure 11, and you could just include a sentence to tell the reader that trajectories ending at 0400 UTC 15th October are qualitatively similar to those ending at 0000 UTC 15th October.
- L417-418: be careful when describing the motion of cold air 'as a density current'; you don't have enough evidence to make that specific claim. Instead you say something like 'in a similar manner to a density current' to be less specific.
- L418-420: not sure what you mean by equilibrium in this discussion. You need to be more specific here and describe the important physical processes. Also, language like '…tends to show that…' is too vague and should be avoided.
- L465: replace "probably" with "likely"
- L486-490: I don't follow the argument. You say that backward trajectory analysis demonstrates that an increasing number of low-level moist air

parcels that have **not** crossed the Albera Massif are found inside convective cells on the lee side of the mountain (i.e. originating on the lee side of the mountain, rather than further S-SE?). You then follow on from that point and say the supply of conditionally unstable air parcels (from the S-SE) along the line is continuous. Have I misunderstood, or do these two sentences contradict each other?

- I like the final paragraph of Section 5, in which you summarise the role of the Albera Massif. Follow-up question: is the topography of the Albera Massif only likely to play an important role when the wind direction is exactly as in this event? If the wind direction was slightly different, would you expect a different region of orography to play a more important role?
- Figure 19: confusing that the direction of the vertical cross section (A → B) in panels (c) and (d) is reversed from that in the virtual potential temperature plots in (a) and (b). It would make sense to reverse the orientation of the x-axis in (c) and (d).
- L530: replace "This probably" with "This difference possibly"
- Figure 20: change the colour scale so that you can more easily distinguish between values around 60 to 80% and those nearer 90% and above.
- The summary paragraph from L588-595 is well-written and nicely set out. However, I don't think the statement that Leslie's remnants are involved in the formation of the cold front CF2 is backed up strongly enough by your analysis. Modify the text to include more evidence supporting this statement, or remove it from the manuscript.

**Technical corrections**

- Model domain notation. Use parent + child rather than father / mother.
- L88: replace with "a small jet branch circumvented the cut-off low to the south"
- L100: "participated"? Not sure of its meaning here.
- L138: "more reflectivities are observed". Change to something like "an extended region of reflectivity > 12 dBZ is observed…"
- Figure 11: equivalent potential temperature above the surface is normally shown in K, not in ºC.
- L452: replace "couples" with "couplets"
- Figure 15 is referenced in the text before Figure 14. Switch the order of the figures, or change the text.
- Brackets need editing in some of your time interval labels in Table 2
- L484-485: "…slightly directional vertical wind shear…" do you mean that there is only a slight change in wind direction with height? Replace "slightly" with "slight" if so.
- L597-598: "Accurately tracking and representing the life cycle…"

---

## Author Comment (AC1) · 11 Jun 2021

The authors thank the referee #1 for the thoughtful and constructive comments. Our response to comments is in the attached pdf supplement.

Please also note the supplement to this comment:
https://wcd.copernicus.org/preprints/wcd-2020-54/wcd-2020-54-AC1-supplement.pdf

---

## Author Comment (AC2) · 11 Jun 2021

**Reply to referees - WCD-2020-54 - "A numerical study to investigate the roles of former hurricane Leslie, orography, and evaporative cooling in the 2018 Aude heavy precipitation event"**

We thank the referees for their thoughtful comments, which we have addressed below. Comments from referees are in *italics* and our response is in upright font. Parts added to the manuscript are in blue and deleted parts are in . Lines refer to the lines of the preprint.

**Reply to anonymous referee #1**

**Generic comments**

*This study investigates the mesoscale dynamics of the heavy precipitation event affecting the Aude region during the heavy precipitation event of the 14-15 October 2018. After a (too ?) long and detailed description, the roles of former hurricane Leslie, the orography, and evaporative cooling are examined. The interaction of the incoming trough and surface cyclone with the orography helped the convective activity focusing west of the quasi-stationary cold front and downwind of the Albera Massif. Leslie's remnants are involved in the formation of the second cold front CF2 and contributed to the supply of low-level conditionally unstable air in the second part of the event. However, the grater contribution to the precipitation over the Aude department mainly originated from moisture coming from the Mediterranean Sea. Finally evaporative cooling did not seem to play a substantial role in the dynamics but only in the control of the total simulated amount of precip. The paper is well written and clearly structured however I feel it sometimes goes too long in the description of the dynamics of the event in a disproportionate way compared to the focus of the research. The analysis of Leslie's contribution and evaporative cooling is resolved in a short part compared to the long introduction on the analysis of the event and the methodology of analysis which however I find accurate. So I do not have a strong recommendation if not to make the article more concise and to the point.*

> As recommended, the article has been made more concise and thus substantially shortened.

**Specific comments**

*I'm doubtful on the expression "personal weather station". What about "private weather stations" belonging to citizen weather observing networks integrating the official Meteo-France Network. Or something similar. I would also put a reference if this initiative is coordinated by the National Met Service.*

> Multiple names are given in the literature to personal weather stations: amateur weather stations (*Bell et al.*, 2013; *Muller et al.*, 2015; *Chapman et al.*, 2017), citizen weather stations (*Meier et al.*, 2017; *Napoly et al.*, 2018; *Nipen et al.*, 2020), personal weather stations (*Muller et al.*, 2015; *Chapman and Bell*, 2018; *McNicholas and Mass*, 2018; *de Vos et al.*, 2020; *Hintz et al.*, 2021), private (automatic) weather stations (*Waller*, 2020). The initiative of using these networks is not coordinated for the moment but Eumetnet is working on it with the recent creation of a working group on crowd-sourcing.
In the abstract, the expression has been detailed: [...] including crowd-sourced observations of personal weather stations.
In the introduction, a reference has been added: [...] from observations of standard and personal weather stations (*Mandement and Caumont*, 2020).

*The literature cited in the introduction is appropriate. I think that the synoptic description, as well the numerical simulations section can be shortened perhaps limiting to the essential level of details needed for the following dynamical assessment.*

> The synoptic description and the numerical simulations section have been shortened.

*I'm also suggesting to shorten the Conclusions eliminating unnecessary details like time of the day references of values of single variables to have a more compact way to present results. In the conclusions bullet points (line 590), I would add the part on the evaporative cooling reported earlier at line 585*

> Conclusion has been shortened and modified as asked.

**Reply to anonymous referee #2**

**Generic comments**

*This study focuses on the investigation of a heavy precipitation event in October 2018 in the Aude region, south-eastern France. Following the investigation by Caumont et al. (2020), there main aspects are evaluated in this*
5 *investigation, the origin of moisture with particular attention paid to the role of the hurricane Leslie, the role of the Pyrenees and the role of the evaporative cooling. Numerical simulations and near-surface observations are examined with this purpose. Given the exceptionality of the event, but the increase in the frequency of such situations, and the enormous fatal consequences, the understanding of this heavy precipitation event is of relevance. The paper is clear and well structured. However, sometimes too descriptive, which could make it difficult for the*
10 *reader to follow the main investigation line of the article. Therefore, I suggest shortening too descriptive sections and focus with more detail in the description of those aspects pointed out as goals of this investigation. The literature cited in the introduction is appropriate although I suggest additionally introducing more recent references, also providing some information and references of previous publications, focusing on the key role of moisture and its origin, topography, and evaporative cooling on heavy precipitation events.*

15 > The article has been substantially shortened, some more recent references have been introduced and some information has been added on previous publications.

**Specific comments**

*1. It could be important to clearly state in the abstract and more in detail in the introduction what is the novelty and contribution of this investigation.*

20 > The abstract and the introduction have been modified to clarify contributions and novelty of this investigation. The main contributions of the study are to investigate, through numerical simulations, the physical processes that led to the location and intensity of the observed rainfall during the 2018 Aude HPE. The realism of the simulated fields is evaluated by comparing them with novel near-surface analyses including personal weather stations.

*2. The model and resolution used for the numerical simulations could be provided already in the abstract.*

25 > As recommended, they have been added in the abstract.
L6: numerical simulations are run at 1 km and 500 m horizontal resolutions. These simulations are  evaluated with

*3. Try to substitute 4.5° in the abstract for a "easier location definition" for the reader.*

> In the abstract, some of the text and the conclusion,  has been replaced by east of the Balearic Islands.

*4. The autumn 2018, and particularly October, was a period in which the western Mediterranean region was*
30 *strongly affected by damaging heavy precipitation events, including the known as the Aude 2018 event. Also, on 9 October 2018 the Balearic Islands suffered catastrophic consequences due to similar heavy precipitation phenomena. Thus, the manuscript will benefit from a description of the conditions in the autumn period for the western Mediterranean, the comparison with the climatological conditions particularly for the north-western Mediterranean, and the description of any possible connection between the two events.*

35 > You're right, the Aude and the Balearic Islands HPEs belong to a series of precipitating episodes that occurred in October and November 2018 as described by *Lorenzo-Lacruz et al.* (2019). For French territories of the western Mediterranean region, Météo-France issued orange or red watches/warnings for thunderstorms, rain or floods for not less than 15 days during the month of October 2018. We agree that a detailed analysis of the large-scale conditions over the north-western Mediterranean region during this autumn period as well as establishing connections between these two HPEs is of particular interest. We think

it is out of the scope of this particular study which focuses on what happened in a short time period at mesoscale but could be the topic of a future work.

A short sentence has been added: This episode is part of a series of HPEs that occurred in October and November 2018 over the north-western Mediterraneannd particularly affected the Balearic Islands on 9 October (*Lorenzo-Lacruz et al.*, 2019) and Italy on 27–30 October (*Davolio et al.*, 2020).

*5. The case description and the numerical simulations section could be shortened with major focus on the information relevant for the case analysis and answering of the questions initially raised.*

> The case description (Sect. 2) and the numerical simulations section (Sect. 3) have been shortened as recommended.

*6. Please rethink the sentence in L215 "... is run to realistically simulate the case".*

>  has been replaced by is carried out to realistically simulate the rainfall observed during the case.

*7. It would be interesting providing an additional simulation in which the Pyrenees were removed since its position is upstream and one goal of this study is provide and analysis of the role of topography. If this is not possible, a hypothesis would be needed.*

> Two additional simulations have been computed in addition with the two simulations (REF and NOALB) already shown in the article. Fig. A summarises the topography of all simulations:

(a) REF: Reference
(b) NOALB: the Albera Massif is removed
(c) NOALCO: the Albera Massif and the Corbières Massif are removed
(d) NOPYR: the Eastern slopes of the Pyrénées, the Corbières Massif and the Albera Massif are removed.

However, because of a change in Météo-France supercomputer (that induced a change in model compilers) and the use of a newer version of Meso-NH due to this supercomputer change (v5.4.4 instead of v5.4.2), the same physical choices have not led to the exact same numerical solution. So, to produce Fig. B, REF and NOALB have also been recomputed with this version 5.4.4. Figures of the article are not modified, they still are from the v5.4.2. The 24 h rainfall of the 4 simulations is shown in Fig. B and can be compared with Fig. 6 of the article. As adding these new figures would increase the length of the article, the results of these additional simulations are explained in a small paragraph:

Additional simulations (not shown) in which the Corbières Massif and the Eastern slopes of the Pyrenees are successively flattened show a substantial decrease in maximum accumulated precipitation downstream of these reliefs, associated with a spread of precipitation above 200 mm over a larger area along the quasi-stationary front.

*8. Please provide information of why only near-surface information is analysed and validated.*

> A comparison to high resolution soundings at the initial state has been added, as well as an explanation of why we focus on the validation of REF near the ground:

Since the initial conditions of REF are provided by the AROME analyses in which all conventional observations are assimilated, there is little deviation from these observations at the initial time. At 12:00 UTC 14 October, comparison of REF fields on all vertical levels to high resolution soundings of Nîmes, Barcelona and Palma (not shown, see Fig. 1 for the locations) reveal absolute bias (respectively root mean square error) of $< 0.3\,\mathrm{K}$ (resp. $< 0.6\,\mathrm{K}$) in temperature, $< 0.3\,\mathrm{g\,kg^{-1}}$ (resp. $< 0.8\,\mathrm{g\,kg^{-1}}$) in water vapour mixing ratio, $< 0.5\,\mathrm{m\,s^{-1}}$ (resp. $< 1.9\,\mathrm{m\,s^{-1}}$) in wind speed and $< 7°$ (resp. $< 16°$) in wind direction.

Because the stationarity of precipitation is correlated to the quasi-stationarity of a MSLP trough and a virtual potential temperature ($\theta_\mathrm{v}$) gradient (Sect. 2.1), and because of the availability of near-surface observations that are not assimilated in the AROME model, this section focuses on validating the REF simulation near the surface.

*9. There are too many figures in the article, and multiple panels in some of them. Please, restrict the number of figures to those strictly necessary and try to combine when possible several information in one figure.*

> Figures 8, 10, 13 have been removed and several figures have been modified as recommended by other referees.

[Figure]

**Figure A.** Orography of the south Languedoc-Roussillon including the Aude department in simulations (a) REF, (b) NOALB, (c) NOALCO and (d) NOPYR.

[Figure]

**Figure B.** 24 h accumulated precipitation between 12:00 UTC 14 October and 12:00 UTC 15 October from the (a) REF (b) NOCOOL, (c) NOALCO and (d) NOPYR simulations. Dotted black lines indicate catchment limits of the Aude basin and its tributaries.

**Reply to anonymous referee #3**

**General comments**

>*Paper would benefit from a more diverse collection of references, which place these heavy rainfall events into wider context and better draw upon literature from all across the world.*

5 > More diverse references have been added to the paper as recommended.

>*A schematic diagram of typical synoptic setup during heavy precipitation event might be useful (you show a map in Figure 1, but as the reader, I want more information and detail). This kind of plot would really sell the paper to the reader.*

> We agree, but given the number of figures this plot have not been added. A schematic diagram of main low-level mechanisms
10 of HPEs in the western Mediterranean region is given by the Fig. 1 of *Ducrocq et al.* (2016), reference which has been added to the article. Also, three composite analyses detailing the synoptic setup for HPEs occurring over Languedoc-Roussillon are given by the Fig. 11 of *Ricard et al.* (2012).

>*The detail in evaluating model performance against observations is impressive. The length of this paper could be reduced. The level of detail is commendable, but there are sections that are difficult to fully take in because of the
15 consistent level of depth.*

>– *The text on the model setup in Section 3.1 is an example. The level of detail is impressive, but do you need to go into this much detail?*
>– *Along the same line, you could lose a couple of figures (and accompanying text) without reducing the quality of your writing. As an example, you could remove either Figure 7 or 8, Figure 10, and Figure 13 without
20 impacting the quality of your analysis.*

> The numerical simulations section (Sect. 3) has been significantly shortened. Details of the Meso-NH configuration necessary to reproduce our experiments have been moved to an appendix. Figures 8, 10, 13 have been removed as recommended and the accompanying text has been shortened.

>*Careful to use the same tense throughout your writing. In places you switch between present and past, which is
25 confusing.*

> You're right, verbs tense has been homogenised throughout the article.

>*When you introduce diagnostics such as water vapour mixing ratio and precipitable water in Section 2, it would be useful to know how large these values are relative to climatology (how unusual?).*

> The sentence introducing water vapour mixing ratio (L98) is detailed: These values are in the upper range of mixing ratios
30 observed within the boundary layer over the Gulf of Lion by *Di Girolamo et al.* (2016) ($8–15\,\mathrm{g\,kg^{-1}}$) or within the free troposphere below 3 km a.g.l. ($2–8\,\mathrm{g\,kg^{-1}}$) over the Balearic Islands by *Chazette et al.* (2016) during southerly marine flows of the HyMeX SOP1.
The sentence referring to precipitable water (PW) L116 has been removed to shorten this part. The 2002-2006 climatology of *Ricard et al.* (2012) shows values of PW during the mature stage of HPEs ranging from 14.1 to 38.4 mm over southern France.
35 *Khodayar et al.* (2018) shows that all convective episodes leading to heavy precipitation systems during IOP12 (2012) occurred in environments with integrated water vapour (IWV) ranging between 30 and 45 $\mathrm{kg\,m^{-2}}$ i.e. PW = $\frac{\mathrm{IWV}}{\rho}$ ranging between 30 and 45 mm considering a water density $\rho = 1000\,\mathrm{kg\,m^{-3}}$.

>*Splitting section 2 into two sub-sections, one focusing on the synoptic evolution and the other on the mesoscale details of the rainfall, could help to make the writing more streamlined and easier to follow. At the moment, there
40 is too much information crammed in; the section is too long.*

> The section has been split in two sub-sections, which have both been shortened.

*You have confused 'westwards' and 'eastwards' in places throughout the text. Make sure that your descriptions are accurate and consistent.*

> You're absolutely right, four errors have been corrected.

  – L85  eastward movement;
  – L86  eastwards;
  – L109  eastwards;
  – L110  east

*There are quite a few occasions where you describe details of the synoptic or mesoscale evolution without referring to figures. It is fine to do this on a few occasions (adding "not shown"), but not too much, or the reader will get confused.*

> Parts of the article which were not related to figures have been shortened or deleted, and "not shown" has been added in sentences not referring to figures.

*Some of the more technical information could be included in an Appendices section at the end of the paper, rather than in the main text (would help to streamline the text). The first paragraph in Section 4 is a good example (on REF_SP).*

> Technical information about Meso-NH configuration have been moved to an appendix. The paragraph in Section 4 has been substantially shortened.

*Generally, there is too much description of results, and not enough interpretation and putting your results into wider context.*

> By shortening the descriptive parts of the article, adding more diverse and recent references to put this HPE in a wider context, and responding to the various referees' comments, we hope to have improved the quality of the article and addressed this point.

*When explaining to the reader why this study is important (Introduction), you should discuss the topic of interactions between tropical cyclones and the midlatitude flow, and how they can impact upon predictability. Papers by Christian Grams and Florian Pantillon provide good examples of this type of work. This type of discussion will help to link the specifics of this event that you're discussing with larger-scale issues of interest in the meteorology community.*

> Thanks for the references. Sentences have been added or modified in the introduction and conclusion based on the information of articles written by Christian Grams and Florian Pantillon:

In the introduction:

Transitioning hurricanes over the North Atlantic are known to disturb the midlatitude flow close or downstream of them, causing or modifying the location and intensity of high-impact weather such as HPEs (*Grams and Blumer*, 2015; *Pantillon et al.*, 2015). As hurricanes can supply large amounts of moisture and because the moisture structure in the lower troposphere was shown to play a key role in the timing and location of precipitation of previous HPEs (*Lee et al.*, 2018), it is of interest to quantify the amount of moisture supplied by Leslie to the convective system.

In the conclusion:

Future work could quantify Leslie's direct contribution in the formation of the Mediterranean low and its associated cold front CF2 but also the remote impact of Leslie's extratropical transition in a similar way as *Grams and Blumer* (2015) or *Pantillon et al.* (2015): it could help to understand how accurately these systems need to be tracked to improve HPE forecasts.

**Specific comments**

*Avoid vague language (a few replacements are suggested)*

- *convective activity -> convection*
- *synoptic situation*
- *seem to be maintained -> maintained*
- *density departures*
- *thermal signature*
- *more reflectivities are observed*
- *precipitating activities -> precipitation*

> Vague language has been replaced in several parts of the manuscript:

L3:  has been replaced by At synoptic scale, the former hurricane Leslie was involved

L14:  has been replaced by are maintained

L44:  has been replaced by buoyancy differences between air masses

L109:  has been replaced by Precipitation along CF1

L110:  has been replaced by thermal gradient

L138:  has been replaced by an extended region of reflectivity >12 dBZ is observed

Some occurrences of "synoptic situation" have been kept when its meaning of "meteorological situation at synoptic-scale" appeared useful.

*Standard and personal weather stations –please elaborate*

> The sentence has been reformulated: with near-surface analyses  including crowd-sourced observations of personal weather stations.

*You introduce the concept of the stationarity of the cold front (L11) without any prior discussion –seems a bit rushed*

> It referred to L4: "At mesoscale, convective cells focused west of a cold front". However, to be consistent with literature (e.g., *Santurette and Joly*, 2002), a cold front that become stationary is a quasi-stationary front. Thus, L4 has been modified: At mesoscale, convective cells focused west of a decaying cold front, that became quasi-stationary,

*L15 to 17: 'the location of the exceptional precipitation appears to be driven primarily by the location of the quasi-stationary cold front...' seems like an obvious statement to make. Would like more insight here.*

> This front is, before the event, a decaying cold front along which precipitation decreased and almost stopped. Then during the event, it becomes active due to the increasing advection of conditionally unstable air by the low-level jet that strengthens at the beginning of the event. More details have been added in the abstract to express the specificity of this quasi-stationary front: At mesoscale, convective cells focused west of a decaying cold front [...] Regarding lifting mechanisms, the advection of conditionally unstable air by a low-level jet towards the quasi-stationary front, confined to altitudes below 2 km, reactivated convection along and downwind of the front.

*Which dataset did Ricard et al. (2012) use for their climatology of heavy precipitation events over the northwest Mediterranean?*

> To answer, the sentence L27 has been expanded: *Ricard et al.* (2012) built a climatology of HPE environments over the north-western Mediterranean area based on 3D-Var ALADIN mesoscale analyses of a 5 yr period (2002–2006).

> *'Mediterranean Sea supplied up to 60% of the total air parcels moisture (Duffourg and Ducrocq, 2013; Duffourg et al. 2018), modulating the intensity of convective precipitation.' How did the authors calculate this value (60%)?*
5  *More information on the method is needed here, even if brief.*

> This value was calculated by water budgets by *Duffourg and Ducrocq* (2013). Also, *Duffourg et al.* (2018) estimated with backward trajectories that 50 % of the moisture supply of the 14 October 2012 HPE originated from evaporation over the Mediterranean sea. Because the method varies between both articles, the last reference has been removed.
The sentence L37 has been modified: The Mediterranean Sea supplies moisture – up to 60 % of the total air parcels moisture in
10  previous HPEs according to water budgets of *Duffourg and Ducrocq* (2013) *(Duffourg and Ducrocq., 2013; Duffourg et al., 2018)*.

> *L46: don't start a new sentence with 'this'. Always refer directly to the part of the previous sentence that you're referencing (i.e. 'This combination of factors...').*

> You're right, it has been corrected. Other sentences L301-302 and L530-531 have also been corrected.

> *L52: need a sentence to tell the reader that you're introducing your case study here. Currently, you just start*
15  *talking and the transition from the topic in the previous paragraph is not smooth enough.*

> The following sentence has been added: Among these mechanisms, those at the origin of the HPE of 14 and 15 October 2018, on which this article focuses, are studied.

> *Paragraph on the event itself reads well (L52-62); you summarise the key points nicely.*

> Thank you!

20  *L68-69: 'Because of the heavy rain observed in the area, evaporative cooling processes may have played a role in the stationarity of the cold front.' Not sure I follow this argument. Evaporative cooling would be expected in conjunction with the rainfall, but why are you hypothesising that this cooling could play an important role in the movement of the front? You need to make this point more confidently here, and with more detail.*

> This point has been detailed: Because of the heavy convective rain observed west of this front, evaporative cooling may have
25  additionally cooled the west side of the front. This additional cold air may have caused a dynamic feedback that contributed to the stationarity of the front. Similar dynamic feedback was described by *Davolio et al.* (2016) over north-eastern Italy: in cases of upstream events, a cold-air layer formation preceded the convection onset and evaporation and sublimation of precipitation beneath the convective system were able to additionally cool this cold-air layer, which influenced the propagation of this cold-air mass.

30  *L69-71: link between evaporative cooling and conditionally unstable air? Are you hypothesising that evaporative cooling occurred in the mid troposphere and destabilised the profile? Need to make these details clearer.*

> You're right, the sentence L69-71 was unclear so it has been reworded. Establishing a link between evaporative cooling and conditionally unstable air was not the goal of the sentence. Consequently, the goal of the article is to address the questions raised by *Caumont et al.* (2021): what are the roles of (i) the moisture provided by Leslie, (ii) the Pyrenees relief and (iii) the
35  evaporative cooling in the physical processes that led to the location and intensity of the observed rainfall?

> *L84-87: where is the evidence that the interaction between both lows seems to have strongly slowed the westward movement of the mid-level cut-off low? Or, do you mean eastward movement? This would make more sense.*

> Indeed, there was an error, we meant eastward. Because it is only speculative, this paragraph has been modified:

- L85  eastward movement;
- L86  eastwards.

*L88-92: which figure are you referring to in this discussion? Make the connection clearer and add detail to figure if necessary.*

5 > There was no figure, (not shown) has been added.

*L103-104: "A potential vorticity at upper levels is observed upstream of the low." Where is the evidence for this feature? Need to relate all statements to figures, or add '(not shown)'.*

> There was indeed no figure. This part has been removed in order to shorten the manuscript as recommended.

*L104: "It may have helped to deepen it...". A reference or two here on the interaction between surface cyclones*
10 *and upper-level PV anomalies would be beneficial. Something like Hoskins et al. (1985), or a classic paper along those lines.*

> This part has been removed in order to shorten the section as recommended.

*Be as precise as possible in your discussion of figures. For example, in Figure 4 you plot the 925 hPa water vapour mixing ratio and geopotential height. In the accompanying text, make sure that you refer to the diagnostics in the*
15 *figure ("925 hPa wind"), rather than using more vague descriptions ("near-surface wind").*

> You're right, here because the 925 hPa wind was not shown in Figure 4, we have added (not shown) and have replaced near-surface by between the surface and 925 hPa.

*Do the authors have any theory on why precipitation within the western band is overestimated by the model, or why the model overestimates precipitation over orography?*

20 > Within the western band and over orography of the eastern Pyrenees, Fig. B shows large departures between quantitative precipitation estimate (QPE) and simulation. However, during that case, because of a radar failure mentioned in the article but also the bad coverage of eastern Pyrenees by the French radar network, QPEs, even if they include rain gauge data, likely underestimate rainfall in these two mountainous areas. Thus, even if direct comparisons of the REF simulation with several rain gauges show overestimations, it is hard to give an explanation as the amplitude of the overestimations away of rain gauges
25 is not known with certainty.

*Add latitude and longitude labels and tickmarks to Figures 2 to 6*

> Labels and tickmarks have been added.

*Overlay the position of key features such as CF1 and CF2 on the relevant figures. It would make the connection between text and figures much stronger.*

30 > It has been done on the relevant figures.

*How do Leslie's remnants contribute to the formation of the cold front CF2, behind which a Mediterranean low deepened rapidly?*

> In the article, we indicate that thermodynamic anomalies associated to Leslie's remnants evolve in what is identified by meteorologists as a cold front on the analyses but the physical mechanisms at stake are not studied and the answer to this
35 question has been mentioned in the conclusion as a future work.

*Mark the Albera Massif on Figure 5. Also mark the Aude and Pyrénées-Orientales departments. You refer to these features, but the reader won't necessarily know where they are (without referring back to earlier figures). The convective lines that you mark on Figure 5 are not easy to see. The eastern line in Figs. 5a to c is fine, but the other two lines are much more difficult to pick out. Is there another way that you could annotate these figure panels?*

5  > In Figure 5, the color range has been modified to better highlight the lines. We have not found a better way than arrows to make clear the direction of propagation of the cells. "A", "P", CF1 and CF2 have been added to respectively indicate the locations of the Albera Massif, the Eastern slopes of the Pyrenees, CF1 and CF2. We have not added the Aude and Pyrénées-Orientales departments as the figure is quite busy and as they are shown in Fig. 1.

*L229: do you mean downstream? Upstream suggests the windward (not lee) side of the ridges.*

10  > Our sentence was unclear: we meant that bands were generated upstream of the ridges, propagated, and then were enhanced downstream. If we quote *Cosma et al.* (2002): "The bands are generated upstream of these ridges and enhanced on the lee side by convergence created by deflection around the obstacle and penetration of the flow into the valleys". Our sentence has been clarified, and the reference to the upstream generation has been removed:
Similar convective band generation was observed in the south-eastern flank of the Massif Central by *Miniscloux et al.* (2001)
15  and *Cosma et al.* (2002)  : rainfall bands were enhanced on the lee side of small-scale topography  ridges.

*L243-255: when justifying your investigation of evaporative cooling, it's not immediately clear how evaporative cooling near the surface could modify the location of CF1 or change its stationarity. Could you describe how this could occur (maybe add a reference)?*

20  > This part has been modified, and a reference has been added in the introduction (see the answer of L68-69).

*L248-255: in the NOCOOL simulation, did you turn evaporation off completely in the child domain, or just set the temperature tendency from evaporation to zero? Assuming the second – it wouldn't make a difference to your results, but it's best to be as precise as possible when discussing the changes you made to the model output.*

> You're right, a sentence is added in the paper: In NOCOOL, negative temperature tendency from evaporation of raindrops is
25  set to zero.

*L277-286: you discuss the evolution of precipitation in your model simulation relative to observations, but don't show any of the plots required to do so (Figure 6 presents accumulated rainfall but gives no indication of how the structure of the rainfall evolves during the 24-h period).*

> You're right, the paragraph has been significantly shortened and (not shown) has been added.

30  *Figure 8: why have you used virtual potential temperature rather than potential temperature to diagnose the position of the front? Would be interesting to know what the pure temperature difference across the front looks like (potential temperature) as well as the moisture difference (virtual potential temperature). Don't change anything in the manuscript, it's just something to think about in future.*

> We used virtual potential temperature because it takes into account the density effects of water vapour. In this case, virtual
35  potential temperature and potential temperature gradients are quite similar because relative humidity is between 90 and 100 % at screen level between 23:00 and 04:00 UTC on each side of the front.

*Figure 10: the idea is good, but it's difficult to pick out the difference between the observations and the REF simulation, because as the reader you are drawn to the filled contours (REF winds) and it's hard to see the differences between the wind vectors in black and red. Given that the text accompanying this figure makes a*
40  *similar point to that with Figure 9, I would recommend removing the figure. You could still make your point about the longer stationary period in the REF simulation (relative to observations) likely explaining some of the over-estimation in accumulated rainfall.*

> You're right, the figure has been removed.

> *L345-346: how does the Lagrangian trajectory tool of Gheusi and Stein (2002) differ from newer tools such as LAGRANTO?*

To answer, the following sentence has been added: It is based on the technique of *Schär and Wernli* (1993) in which three Eulerian passive tracers are initialised with the initial grid point position and are advected online by the resolved and subgrid-scale wind; a review of existing Lagrangian trajectory tools is given by *Miltenberger et al.* (2013).

> *Discussion of Figure 11 ( L355): make it clearer in the figure panels and in the text that the trajectories in (b) correspond to the transect (A -> D), etc. The reader will get confused otherwise. You have done this in your discussion of Figure 13 please apply the same method to the discussion of Figure 11.*

> The transects have been indicated in the text as recommended.

> *Nice illustration of lifting along the cold front in Fig. 11c. Overlay the position of the cold fronts (CF1 and CF2) in Figures 11 and 12, and on Figures 14 and 15.*

> The position of the cold fronts has been overlaid.

> *L387-389: the claim that the trough over the Aude department may locally increase wind shear inland, based on Fig. 10a, is not based up strongly by the evidence.*

> You're right, this claim has been removed.

> *Do you need Figure 13? The results are similar to those in Figure 11, and you could just include a sentence to tell the reader that trajectories ending at 0400 UTC 15th October are qualitatively similar to those ending at 0000 UTC 15th October.*

> Figure 13 has been deleted, and a sentence has been added: Similar backward trajectories are simulated at the end of this first part, e.g. at 04:00 UTC 15 October (not shown).

> *L417-418: be careful when describing the motion of cold air 'as a density current'; you don't have enough evidence to make that specific claim. Instead you say something like 'in a similar manner to a density current' to be less specific.*

> *L418-420: not sure what you mean by equilibrium in this discussion. You need to be more specific here and describe the important physical processes. Also, language like '...tends to show that...' is too vague and should be avoided.*

> "As a density current" has been replaced by "in a similar manner to a density current". The expression "tends to show that" has been removed. Here equilibrium referred to the fact that the propagation of the cold air located west of the quasi-stationary front towards east appears countered by the low-level jet blowing perpendicularly to it.
This part has been rewritten: ~~When wind speed decreased along CF1, the cold air west of CF1 started flowing eastwards, towards the Mediterranean Sea, as a density current, spreading out circularly over the sea. This cold air rapidly flowing as soon as the LLJ stops tends to show that an equilibrium maintaining CF1 quasi-stationary was reached between CF1 and the LLJ. Then, the equilibrium progressively broke from south to north by the advance of CF2.~~ When wind speed abruptly decreases along CF1, CF1 stationarity breaks and the cold air west of CF1 immediately starts flowing eastwards in a similar manner to a density current, and later spreads out circularly over the Mediterranean Sea (not shown). It indicates that, during the HPE, the propagation of the cold air located west of CF1 is countered by the LLJ blowing perpendicularly to it, also in a similar manner as the propagation of a cold pool can be countered by the environmental wind (*Miglietta and Rotunno*, 2014).

> *L465: replace "probably" with "likely"*

> The modification has been made.

*L486-490: I don't follow the argument. You say that backward trajectory analysis demonstrates that an increasing number of low-level moist air parcels that have not crossed the Albera Massif are found inside convective cells on the lee side of the mountain (i.e. originating on the lee side of the mountain, rather than further S-SE?). You then follow on from that point and say the supply of conditionally unstable air parcels (from the S-SE) along the line is continuous. Have I misunderstood, or do these two sentences contradict each other?*

> The sentences were unclear. Parcels that do not cross the Albera Massif (which is at the S-SE) are carried by the south-eastern LLJ and originate from the Mediterranean Sea, not from the lee side of the mountain. This part has been rewritten:
Because of the slight directional vertical wind shear simulated in the lower part of the troposphere (see wind direction in Figs. 17a,c), ~~the LLJ continuously supplies conditionally unstable air parcels to the convective cells formed from their southeastern flank while these cells are advected by the south-southeastern mid-level wind. Backward trajectories starting from their updrafts (not shown) show that once convective cells are on the lee side of the mountain, as they are advected north-northwestwards, an increasing number of low-level moist air parcels that have not crossed the Albera Massif are found inside the cells. Thus, the supply of conditionally unstable air parcels along the line is continuous and possibly explains the maintenance of the convective cells long after they are formed.~~

once convective cells are on the lee side of the mountain, as they are advected by the south–south-eastern mid-level wind, the south-eastern LLJ supplies conditionally unstable air parcels that do not cross the Albera Massif to the cells from their south-eastern flank. Backward trajectories starting from their updraughts (not shown) indicate that the number of low-level moist air parcels that do not cross the Albera Massif found inside the cells increases as they are advected. This supply mechanism possibly explains the maintenance of the convective cells long after they are formed.

20

*I like the final paragraph of Section 5, in which you summarise the role of the Albera Massif. Follow-up question: is the topography of the Albera Massif only likely to play an important role when the wind direction is exactly as in this event? If the wind direction was slightly different, would you expect a different region of orography to play a more important role?*

25  > Thank you!
Considering a Mediterranean air mass having a conditional instability similar to this case, if the wind direction was slightly different and we are talking about the role on rainfall, we expect that the Albera Massif could still play an important role: Fig. 9 of *Ducrocq et al.* (2008) shows that the Albera Massif was a source of lifting in the 1999 Aude HPE (see the parcel n°1) while the 500 hPa wind direction (Fig. 18 of *Nuissier et al.*, 2008) was slightly further south than in 2018 (with also differences in 30 wind speed). Depending on the amplitude of wind direction changes, we indeed expect (i) a different region of orography to play a more important role (ii) different interactions with other reliefs: e.g. if the mid-level wind had an easterly direction, convective cells would be advected from the Albera Massif towards the highest reliefs of the Pyrenees (iii) different wind shear between the ground and mid-levels, wind shear that may not be as favourable as in this 2018 case to the maintenance of convective cells downstream of the Albera Massif.

35  *Figure 19: confusing that the direction of the vertical cross section (A -> B) in panels (c) and (d) is reversed from that in the virtual potential temperature plots in (a) and (b). It would make sense to reverse the orientation of the x-axis in (c) and (d).*

> You're right, the orientation of the cross-section has been reversed in Fig. 19 and corresponding changes in the text have been made.

40  *L530: replace "This probably" with "This difference possibly"*

> The modification has been made.

*Figure 20: change the colour scale so that you can more easily distinguish between values around 60 to 80% and those nearer 90% and above.*

> The colour scale has been changed.

*The summary paragraph from L588-595 is well-written and nicely set out. However, I don't think the statement that Leslie's remnants are involved in the formation of the cold front CF2 is backed up strongly enough by your analysis. Modify the text to include more evidence supporting this statement, or remove it from the manuscript.*

> Thanks! You're right, the fact that Leslie's remnants are involved in the formation of the cold front CF2 has been moved to future work.

**Technical corrections**

*Model domain notation. Use parent + child rather than father / mother.*

> You're right,  L226 has been replaced by parent.

*L88: replace with "a small jet branch circumvented the cut-off low to the south"*

> It has been replaced.

*L100: "participated"? Not sure of its meaning here.*

> It has been replaced by "contributed".

*L138: "more reflectivities are observed". Change to something like "an extended region of reflectivity > 12 dBZ is observed..."*

> Done,  has been replaced by an extended region of reflectivity >12 dBZ is observed

*Figure 11: equivalent potential temperature above the surface is normally shown in K, not in °C.*

> The modification has been made.

*L452: replace "couples" with "couplets"*

> The modification has been made.

*Figure 15 is referenced in the text before Figure 14. Switch the order of the figures, or change the text.*

> The order of figures has been switched.

*Brackets need editing in some of your time interval labels in Table 2*

> Hyphens have been replaced by commas to match standard interval notations. The first seven intervals are voluntarily right-open as well as the last one is closed.

*L484-485: "...slightly directional vertical wind shear..." do you mean that there is only a slight change in wind direction with height? Replace "slightly" with "slight" if so.*

> You're right, it has been corrected.

*L597-598: "Accurately tracking and representing the life cycle..."*

> It has been corrected.

**References**

Bell, S., D. Cornford, and L. Bastin, The state of automated amateur weather observations, *Weather*, *68*(2), 36–41, https://doi.org/10.1002/wea.1980, 2013.

Caumont, O., M. Mandement, F. Bouttier, J. Eeckman, C. Lebeaupin Brossier, A. Lovat, O. Nuissier, and O. Laurantin, The heavy precipitation event of 14–15 october 2018 in the aude catchment: a meteorological study based on operational numerical weather prediction systems and standard and personal observations, *Natural Hazards and Earth System Sciences*, *21*(3), 1135–1157, https://doi.org/10.5194/nhess-21-1135-2021, 2021.

Chapman, L., and S. J. Bell, High-Resolution Monitoring of Weather Impacts on Infrastructure Networks Using the Internet of Things, *Bulletin of the American Meteorological Society*, *99*(6), 1147–1154, https://doi.org/10.1175/BAMS-D-17-0214.1, 2018.

Chapman, L., C. Bell, and S. Bell, Can the crowdsourcing data paradigm take atmospheric science to a new level? A case study of the urban heat island of London quantified using Netatmo weather stations, *International Journal of Climatology*, *37*(9), 3597–3605, https://doi.org/10.1002/joc.4940, 2017.

Chazette, P., et al., A multi-instrument and multi-model assessment of atmospheric moisture variability over the western mediterranean during hymex, *Quarterly Journal of the Royal Meteorological Society*, *142*(S1), 7–22, https://doi.org/10.1002/qj.2671, 2016.

Cosma, S., E. Richard, and F. Miniscloux, The role of small-scale orographic features in the spatial distribution of precipitation, *Quarterly Journal of the Royal Meteorological Society*, *128*(579), 75–92, https://doi.org/10.1256/00359000260498798, 2002.

Davolio, S., A. Volonté, A. Manzato, A. Pucillo, A. Cicogna, and M. E. Ferrario, Mechanisms producing different precipitation patterns over north-eastern italy: insights from hymex-sop1 and previous events, *Quarterly Journal of the Royal Meteorological Society*, *142*(S1), 188–205, https://doi.org/10.1002/qj.2731, 2016.

Davolio, S., S. D. Fera, S. Laviola, M. M. Miglietta, and V. Levizzani, Heavy precipitation over italy from the mediterranean storm "vaia" in october 2018: Assessing the role of an atmospheric river, *Monthly Weather Review*, *148*(9), 3571 – 3588, https://doi.org/10.1175/MWR-D-20-0021.1, 2020.

de Vos, L. W., A. M. Droste, M. J. Zander, A. Overeem, H. Leijnse, B. G. Heusinkveld, G. J. Steeneveld, and R. Uijlenhoet, Hydrometeorological monitoring using opportunistic sensing networks in the Amsterdam metropolitan area, *Bulletin of the American Meteorological Society*, *101*(2), E167–E185, https://doi.org/10.1175/BAMS-D-19-0091.1, 2020.

Di Girolamo, P., C. Flamant, M. Cacciani, E. Richard, V. Ducrocq, D. Summa, D. Stelitano, N. Fourrié, and F. Saïd, Observation of low-level wind reversals in the gulf of lion area and their impact on the water vapour variability, *Quarterly Journal of the Royal Meteorological Society*, *142*(S1), 153–172, https://doi.org/10.1002/qj.2767, 2016.

Ducrocq, V., O. Nuissier, D. Ricard, C. Lebeaupin, and T. Thouvenin, A numerical study of three catastrophic precipitating events over southern France. II: Mesoscale triggering and stationarity factors, *Quarterly Journal of the Royal Meteorological Society*, *134*(630), 131–145, https://doi.org/10.1002/qj.199, 2008.

Ducrocq, V., S. Davolio, R. Ferretti, C. Flamant, V. H. Santaner, N. Kalthoff, E. Richard, and H. Wernli, Introduction to the hymex special issue on 'advances in understanding and forecasting of heavy precipitation in the mediterranean through the hymex sop1 field campaign', *Quarterly Journal of the Royal Meteorological Society*, *142*(S1), 1–6, https://doi.org/10.1002/qj.2856, 2016.

Duffourg, F., and V. Ducrocq, Assessment of the water supply to Mediterranean heavy precipitation: a method based on finely designed water budgets, *Atmospheric Science Letters*, *14*(3), 133–138, https://doi.org/10.1002/asl2.429, 2013.

Duffourg, F., K.-O. Lee, V. Ducrocq, C. Flamant, P. Chazette, and P. Di Girolamo, Role of moisture patterns in the backbuilding formation of HyMeX IOP13 heavy precipitation systems, *Quarterly Journal of the Royal Meteorological Society*, *144*(710), 291–303, https://doi.org/10.1002/qj.3201, 2018.

Grams, C. M., and S. R. Blumer, European high-impact weather caused by the downstream response to the extratropical transition of north atlantic hurricane katia (2011), *Geophysical Research Letters*, *42*(20), 8738–8748, https://doi.org/10.1002/2015GL066253, 2015.

Hintz, K. S., C. McNicholas, R. Randriamampianina, H. T. P. Williams, B. Macpherson, M. Mittermaier, J. Onvlee-Hooimeijer, and B. Szintai, Crowd-sourced observations for short-range numerical weather prediction: Report from ewglam/srnwp meeting 2019, *Atmospheric Science Letters*, *n/a*(n/a), e1031, https://doi.org/10.1002/asl.1031, 2021.

Khodayar, S., B. Czajka, A. Caldas-Alvarez, S. Helgert, C. Flamant, P. Di Girolamo, O. Bock, and P. Chazette, Multi-scale observations of atmospheric moisture variability in relation to heavy precipitating systems in the northwestern mediterranean during hymex iop12, *Quarterly Journal of the Royal Meteorological Society*, *144*(717), 2761–2780, https://doi.org/10.1002/qj.3402, 2018.

Lee, K.-O., C. Flamant, F. Duffourg, V. Ducrocq, and J.-P. Chaboureau, Impact of upstream moisture structure on a back-building convective precipitation system in south-eastern france during hymex iop13, *Atmospheric Chemistry and Physics*, *18*(23), 16,845–16,862, https://doi.org/10.5194/acp-18-16845-2018, 2018.

Lorenzo-Lacruz, J., A. Amengual, C. Garcia, E. Morán-Tejeda, V. Homar, A. Maimó-Far, A. Hermoso, C. Ramis, and R. Romero, Hydro-meteorological reconstruction and geomorphological impact assessment of the October 2018 catastrophic flash flood at Sant Llorenç, Mallorca (Spain), *Natural Hazards and Earth System Sciences*, *19*(11), 2597–2617, https://doi.org/10.5194/nhess-19-2597-2019, 2019.

Mandement, M., and O. Caumont, Contribution of personal weather stations to the observation of deep-convection features near the ground, *Natural Hazards and Earth System Sciences*, *20*(1), 299–322, https://doi.org/10.5194/nhess-20-299-2020, 2020.

McNicholas, C., and C. F. Mass, Smartphone pressure collection and bias correction using machine learning, *Journal of Atmospheric and Oceanic Technology*, *35*(3), 523–540, https://doi.org/10.1175/JTECH-D-17-0096.1, 2018.

Meier, F., D. Fenner, T. Grassmann, M. Otto, and D. Scherer, Crowdsourcing air temperature from citizen weather stations for urban climate research, *Urban Climate*, *19*, 170 – 191, https://doi.org/10.1016/j.uclim.2017.01.006, 2017.

Miglietta, M. M., and R. Rotunno, Numerical simulations of sheared conditionally unstable flows over a mountain ridge, *Journal of the Atmospheric Sciences*, *71*(5), 1747–1762, https://doi.org/10.1175/JAS-D-13-0297.1, 2014.

Miltenberger, A. K., S. Pfahl, and H. Wernli, An online trajectory module (version 1.0) for the nonhydrostatic numerical weather prediction model cosmo, *Geoscientific Model Development*, *6*(6), 1989–2004, https://doi.org/10.5194/gmd-6-1989-2013, 2013.

Miniscloux, F., J. D. Creutin, and S. Anquetin, Geostatistical analysis of orographic rainbands, *Journal of Applied Meteorology*, *40*(11), 1835–1854, https://doi.org/10.1175/1520-0450(2001)040<1835:GAOOR>2.0.CO;2, 2001.

Muller, C., L. Chapman, S. Johnston, C. Kidd, S. Illingworth, G. Foody, A. Overeem, and R. Leigh, Crowdsourcing for climate and atmospheric sciences: current status and future potential, *International Journal of Climatology*, *35*(11), 3185–3203, https://doi.org/10.1002/joc.4210, 2015.

Napoly, A., T. Grassmann, F. Meier, and D. Fenner, Development and application of a statistically-based quality control for crowdsourced air temperature data, *Frontiers in Earth Science*, *6*, 118, https://doi.org/10.3389/feart.2018.00118, 2018.

Nipen, T. N., I. A. Seierstad, C. Lussana, J. Kristiansen, and Ø. Hov, Adopting Citizen Observations in Operational Weather Prediction, *Bulletin of the American Meteorological Society*, *101*(1), E43–E57, https://doi.org/10.1175/BAMS-D-18-0237.1, 2020.

Nuissier, O., V. Ducrocq, D. Ricard, C. Lebeaupin, and S. Anquetin, A numerical study of three catastrophic precipitating events over southern France. I: Numerical framework and synoptic ingredients, *Quarterly Journal of the Royal Meteorological Society*, *134*(630), 111–130, https://doi.org/10.1002/qj.200, 2008.

Pantillon, F., J.-P. Chaboureau, and E. Richard, Remote impact of north atlantic hurricanes on the mediterranean during episodes of intense rainfall in autumn 2012, *Quarterly Journal of the Royal Meteorological Society*, *141*(688), 967–978, https://doi.org/10.1002/qj.2419, 2015.

Ricard, D., V. Ducrocq, and L. Auger, A Climatology of the Mesoscale Environment Associated with Heavily Precipitating Events over a Northwestern Mediterranean Area, *Journal of Applied Meteorology and Climatology*, *51*(3), 468–488, https://doi.org/10.1175/JAMC-D-11-017.1, 2012.

Santurette, P., and A. Joly, ANASYG/PRESYG, Météo-France's new graphical summary of the synoptic situation, *Meteorological Applications*, *9*(2), 129–154, https://doi.org/10.1017/S1350482702002013, 2002.

Schär, C., and H. Wernli, Structure and evolution of an isolated semi-geostrophic cyclone, *Quarterly Journal of the Royal Meteorological Society*, *119*(509), 57–90, https://doi.org/10.1002/qj.49711950904, 1993.

Waller, J. A., Editorial: The use of unconventional observations in numerical weather prediction, *Meteorological Applications*, *27*(5), e1948, https://doi.org/10.1002/met.1948, 2020.

---

## Editor Comment (EC1) · Shira Raveh-Rubin (Editor) · 13 Jun 2021

Dear Marc and Olivier,

Thank you for submitting your detailed response to the reviewers' comments.

I invite you to submit a revised manuscript, which will then be sent back to the reviewers for consideration. Please address the concern by all reviewers about the too-long length of the paper, and shorten the text and the number of figures accordingly. In
addition, please provide a wider context to the three goals of the paper in the introduction and to the results, as suggested. The overall guiding thought is to improve the manuscript's readability by focusing on the three goals of the paper and reducing the level of details for issues that divert from the main goals or were addressed in Caumont et al. (2020).

Looking forward to receiving your revised manuscript,

Shira

---

## Referee Report (RR1)

**Reviewer #3**

**Paper**: A numerical study to investigate the roles of former hurricane Leslie, orography, and evaporative cooling in the 2018 Aude heavy precipitation event (R1)

**Authors:** Marc Mandement and Olivier Caumont

**Overview**

- The authors have addressed all referee comments thoroughly, and the paper is much improved with a much better balance between description of the results, and interpretation of the results including the wider context. I am happy to recommend acceptance of this paper once the remaining, small comments have been addressed. Well done on an interesting, informative and well-presented piece of research!

**General comments**

- Replace 'resolution' with 'grid spacing' when using numbers to discuss your model setup (e.g. 1 km, 500 m).
- The length of the paper has been reduced nicely without any loss of quality. Well done!
- You have substantially improved the readability of the manuscript by shortening the description of your results while interpreting your results in more detail.
- The connection between the text and the figures is now stronger. Adding annotations to the figures has helped in this regard.

**Specific comments**

Abstract

- L7: 1 km and 500 m horizontal grid spacing

Introduction

- Refer directly to one of the schematic diagrams of heavy precipitation events that you listed in your previous response (e.g. Fig. 1 of Ducrocq *et al.* 2016; Fig. 11 of Ricard *et al.* 2012), rather that just the paper. This way, the reader won't be expecting you to produce a schematic diagram of your own.
- L30. 'a maritime part of the Occitanie region…'.
- L38. 'abnormally warm SSTs…'.
- L44-45. Couple of other references for outflow boundaries of cold pools, local convergence lines and mesoscale pressure troughs?
- L79. 'A similar dynamic feedback…'.
- Spell out 'Section' fully, rather than using 'Sect'.

Case description

- I can't see the labels "A" and "P" on Figure 5, even though you refer to them in the figure caption.
- L146. Include a couple of references for the sentence on the catastrophic consequences of the rainfall (from the earlier list on L65-66).

**Origin of the conditionally unstable air and lifting mechanisms**

- L251. Replace 'what' with 'which'.
- L283. Delete 'brutally'.
- L287. Replace 'increase' with 'increases'.
- L293. '…over the sea'.
- L309. '…local forcing'.
- L310. 'Convection triggered over the sea…'
- L388. Instead of 'south–south-eastern wind', use 'south south-easterly wind'. Do the same for any other instances throughout the paper.
- L387-390. The reworked sentence is slightly confusing to read. Can you reword by splitting into two sentences, or changing the order of the points you're making?
- L395. Reword to avoid starting the sentence with an abbreviation ('REF').
- L406. Replace 'relief' with 'peak'.

**Influence of the cooling associated with the evaporation of precipitation**

- L424-425. '…explained by the evaporative cooling being switched off.'

**Conclusions**

- L434-435. Although you have added a couple of sentences at the end of this section addressing Leslie's role as part of a discussion on future work, you still also have a sentence here where you indicate that Leslie's remnants are involved in the formation of the surface low and cold front (CF2). Did you not mean to remove this sentence?

---

## Author Response (AR2)

**Reply to referees - WCD-2020-54 - "A numerical study to investigate the roles of former hurricane Leslie, orography, and evaporative cooling in the 2018 Aude heavy precipitation event"**

We thank the referee #3 for its thoughtful comments on the revised version, which we have addressed below. Comments from the referee #3 are in *italics* and our response is in upright font. Parts added to the manuscript are in blue and deleted parts are in . Lines refer to the lines of the version commented by the referee.

**Reply to anonymous referee #3**

*Replace 'resolution' with 'grid spacing' when using numbers to discuss your model setup (e.g. 1 km, 500 m). L7: 1 km and 500 m horizontal grid spacing*

> Several occurrences have been replaced:

L7: [...] numerical simulations are run at 1 km and 500 m horizontal  grid spacing and evaluated [...]

L141: [...] ANTILOPE quantitative precipitation estimate (QPE) at 1 km horizontal  grid spacing blending [...]

L154-156: A $960 \times 900\,\text{km}^2$ horizontal domain  with a 1 km  grid spacing covering [...] and a $180 \times 135\,\text{km}^2$ horizontal domain  with a 500 m  grid spacing centred over Aude is chosen for the child domain (Fig. 1).

L208: is close to the horizontal  grid spacing of REF.

L441: [...] numerical simulation at 1 km and 500 m horizontal  grid spacing is carried out [...]

L506: For the child model  with a horizontal grid spacing of 500 m, [...]

L508: At  horizontal grid spacing lower or equal to 1 km, [...]

*Refer directly to one of the schematic diagrams of heavy precipitation events that you listed in your previous response (e.g. Fig. 1 of Ducrocq et al. 2016; Fig. 11 of Ricard et al. 2012), rather that just the paper. This way, the reader won't be expecting you to produce a schematic diagram of your own.*

*L30. 'a maritime part of the Occitanie region...'.*

> References have been added and corrections have been made.

With this climatology, synoptic situations favouring HPEs over Languedoc-Roussillon, the maritime part of the Occitanie region in southern France (Fig. 1), are now well known  (e.g. Fig. 11 of *Ricard et al.* 2012; Fig. 1 of *Ducrocq et al.* 2016).

*L38. 'abnormally warm SSTs...'.*

> It has been corrected.

[...] abnormally warm SSTs [...]

*L44-45. Couple of other references for outflow boundaries of cold pools, local convergence lines and mesoscale pressure troughs?*

> New references have been added. Mesoscale pressure troughs have been removed from the enumeration because they are often collocated with convergence lines and we have not found a recent reference supporting this claim (in France for example, pressure is only observed at a synoptic scale by conventional networks).

Such stationary boundaries can be fronts (*Trapero et al.*, 2013), outflow boundaries of cold pools (*Ducrocq et al.*, 2008), local convergence lines (*Buzzi et al.*, 2014) , among others.

*L79. 'A similar dynamic feedback...'.*

> It has been corrected.

 A similar dynamic feedback [...]

*Spell out 'Section' fully, rather than using 'Sect'.*

> It has been corrected.
Section

*I can't see the labels "A" and "P" on Figure 5, even though you refer to them in the figure caption.*

5  > White boxes have been added to improve the readability of labels. Also, the dashed line showing CF1's position has been replaced by the standard symbol of a quasi-stationary front.

*L146. Include a couple of references for the sentence on the catastrophic consequences of the rainfall (from the earlier list on L65-66).*

> Two references have been added.
10  [...] because most of the rain fell in 6 to 12 h (*Préfecture de l'Aude*, 2018; *Ayphassorho et al.*, 2019).

*L251. Replace 'what' with 'which'.*

> It has been replaced.
This section investigates  which mechanisms supply [...]

*L283. Delete 'brutally'.*

15  > It has been deleted.
[...] horizontal wind speed  decreases [...]

*L287. Replace 'increase' with 'increases'.*

> It has been replaced. "Mean" has also been added to be more precise.
Their  mean water vapour mixing ratio increases through [...]

20  *L293. '...over the sea'.*

> It has been corrected.
[...] over the sea.

*L309. '...local forcing'.*

> It has been corrected.
25  [...] local forcing [...]

*L310. 'Convection triggered over the sea...'*

> It has been corrected.
Convection triggered over the sea [...]

*L388. Instead of 'south–south-eastern wind', use 'south south-easterly wind'. Do the same for any other instances*
30  *throughout the paper.*

> You're right, all instances have been corrected.
L236: [...] simulates  south-easterly instead of easterly wind directions in some places.
L279: the speed of this  south-easterly LLJ exceeds
L356: [...] showing that convective cells are aligned with the  south–south-easterly mid-level wind di-
35  rection.
L388-389: [...] as they are advected by the  south–south-easterly mid-level wind  (Fig. 14a), the south-easterly LLJ (Fig. 14c) supplies

*L387-390. The reworked sentence is slightly confusing to read. Can you reword by splitting into two sentences, or changing the order of the points you're making?*

> The sentence has been split and reworded.
 Once convective cells are on the lee side of the mountain, as they are advected by the  south–south-easterly mid-level wind  (Fig. 14a), the south-easterly LLJ (Fig. 14c) supplies conditionally unstable air parcels that do not cross the Albera Massif to the cells from their south-eastern flank. Backward trajectories starting from their updraughts (not shown) indicate that the number of low-level moist air parcels that do not cross the Albera Massif found inside the cells increases as they are advected. This  slight directional vertical wind shear simulated in the lower part of the troposphere possibly explains the maintenance of the convective cells long after they are formed.

*L395. Reword to avoid starting the sentence with an abbreviation ('REF').*

> The sentence has been reworded.
 Maximum precipitation over plains is reduced from 338 mm in REF to 310 mm in NOALB, and the maximum in NOALB (332 mm) is shifted over mountains.

*L406. Replace 'relief' with 'peak'.*

> It has been replaced.
[...] of these peaks, [...]

*L424-425. '...explained by the evaporative cooling being switched off.'*

> It has been corrected.
[...] by the evaporative cooling being switched off.

*L434-435. Although you have added a couple of sentences at the end of this section addressing Leslie's role as part of a discussion on future work, you still also have a sentence here where you indicate that Leslie's remnants are involved in the formation of the surface low and cold front (CF2). Did you not mean to remove this sentence?*

> You're right. The sentence has been removed and the first paragraph has been adapted accordingly.
The synoptic situation on 14 and 15 October 2018 was favourable to a HPE over Languedoc-Roussillon. The rapid deepening of a Mediterranean surface low, extended by a trough over Languedoc-Roussillon, contributed to strengthen a low-level jet (LLJ) over the Mediterranean Sea. Meanwhile, a decaying cold front (CF1) remained quasi-stationary in the middle of the Aude department, west of the trough. The slow movement northwards of the surface low and its associated cold front (CF2) as well as the quasi-stationarity of the trough sustained quasi-stationary atmospheric conditions that continuously supplied conditionally unstable air parcels during several hours over the Aude department.

**References**

Ayphassorho, H., G. Pipien, I. Guion de Meritens, and D. Lacroix, Retour d'expérience des inondations du 14 au 17 octobre 2018 dans l'Aude, https://cgedd.documentation.developpement-durable.gouv.fr/notice?id=Affaires-0011552&reqId= 548839ae-f685-4c3d-bbdb-9452154f508c&pos=12 (last access: 15 October 2020), 2019.

5 Buzzi, A., S. Davolio, P. Malguzzi, O. Drofa, and D. Mastrangelo, Heavy rainfall episodes over liguria in autumn 2011: numerical forecasting experiments, *Natural Hazards and Earth System Sciences*, *14*(5), 1325–1340, https://doi.org/10.5194/nhess-14-1325-2014, 2014.

Ducrocq, V., O. Nuissier, D. Ricard, C. Lebeaupin, and T. Thouvenin, A numerical study of three catastrophic precipitating events over southern France. II: Mesoscale triggering and stationarity factors, *Quarterly Journal of the Royal Meteorological Society*, *134*(630), 131–145, https://doi.org/10.1002/qj.199, 2008.

10 Ducrocq, V., S. Davolio, R. Ferretti, C. Flamant, V. H. Santaner, N. Kalthoff, E. Richard, and H. Wernli, Introduction to the hymex special issue on 'advances in understanding and forecasting of heavy precipitation in the mediterranean through the hymex sop1 field campaign', *Quarterly Journal of the Royal Meteorological Society*, *142*(S1), 1–6, https://doi.org/10.1002/qj.2856, 2016.

Préfecture de l'Aude, Communiqué de presse du 17 octobre 2018, http://www.aude.gouv.fr/IMG/pdf/20181017_cp_21h00.pdf (last access: 15 October 2020), 2018.

15 Ricard, D., V. Ducrocq, and L. Auger, A Climatology of the Mesoscale Environment Associated with Heavily Precipitating Events over a Northwestern Mediterranean Area, *Journal of Applied Meteorology and Climatology*, *51*(3), 468–488, https://doi.org/10.1175/JAMC-D-11-017.1, 2012.

Trapero, L., J. Bech, and J. Lorente, Numerical modelling of heavy precipitation events over eastern pyrenees: Analysis of orographic effects, *Atmospheric Research*, *123*, 368–383, https://doi.org/10.1016/j.atmosres.2012.09.014, 6th European Conference on Severe Storms 2011.

20 Palma de Mallorca, Spain, 2013.